# Preoperative Prognostic Score for Patients with Intrahepatic Cholangiocarcinoma Undergoing Curative-Intent Resection

**DOI:** 10.3390/medsci14010023

**Published:** 2026-01-05

**Authors:** Jarin Chindaprasirt, Thanachai Sanlung, Piyakarn Watcharenwong, Vasin Thanasukarn, Apiwat Jareanrat, Natcha Khuntikeo, Tharatip Srisuk, Prakasit Sa-Ngiamwibool, Chaiwat Aphivatanasiri, Watcharin Loilome, Piya Prajumwongs, Attapol Titapun

**Affiliations:** 1Medical Oncology Program, Department of Medicine Srinagarind Hospital, Khon Kaen University, Khon Kaen 40002, Thailand; 2Cholangiocarcinoma Research Institute, Khon Kaen University, Khon Kaen 40002, Thailand; 3Department of Surgery, Faculty of Medicine, Khon Kaen University, Khon Kaen 40002, Thailand; 4Department of Pathology, Faculty of Medicine, Khon Kaen University, Khon Kaen 40002, Thailand; 5Systems Biosciences and Computational Medicine, Faculty of Medicine, Khon Kaen University, Khon Kaen 40002, Thailand

**Keywords:** intrahepatic cholangiocarcinoma, preoperative score, prognosis, disease-free survival, overall survival

## Abstract

Background: Preoperative inflammatory and nutrition-related markers—including the neutrophil-to-lymphocyte ratio (NLR), lymphocyte-to-monocyte ratio (LMR), prognostic nutritional index (PNI), and controlling nutritional status (CONUT) score—have shown prognostic relevance in various malignancies. However, their comparative utility in predicting recurrence and survival across clinically relevant subgroups in patients with intrahepatic cholangiocarcinoma (iCCA) undergoing curative-intent resection remains unclear. Methods: This retrospective study included 213 patients with histologically confirmed iCCA who underwent curative-intent resection between 2015 and 2021. Preoperative NLR, LMR, PNI, and CONUT scores were calculated from laboratory data obtained within one week before resection. Clinicopathological variables, recurrence, and survival outcomes were analyzed using Cox regression and Kaplan–Meier methods. Results: A preoperative NLR ≥ 2.4 was independently associated with poorer DFS (HR = 1.66, *p* = 0.025) and OS (HR = 1.94, *p* = 0.006). This effect remained significant in patients with R0 resection (DFS: HR = 1.66, *p* = 0.004; OS: HR = 2.11, *p* = 0.014) and in those who subsequently developed recurrence (OS: HR = 1.83, *p* = 0.004). The CONUT score was correlated with OS in both R0 and recurrent subgroups. Tumor morphology, consistent with prior reports, was identified as a postoperative pathological factor associated with worse prognosis. Conclusions: Preoperative NLR was associated with poorer DFS and OS in iCCA patients undergoing curative-intent resection. This association was consistently observed in subgroups with R0 resection and in those who developed recurrence. Meanwhile, the CONUT score showed limited independent significance only among patients with R0 resection who experienced recurrence.

## 1. Introduction

Intrahepatic cholangiocarcinoma (iCCA), a malignancy arising from the epithelial lining of the intrahepatic bile ducts, is the second most common primary liver cancer and accounts for approximately 10–20% of all hepatic malignancies [1,2]. Globally, the incidence of iCCA has been increasing, with the highest rates observed in Southeast Asia, particularly in northeastern Thailand, where chronic liver fluke (*Opisthorchis viverrini*) infection is a well-established carcinogenic factor [3]. The incidence of cholangiocarcinoma (CCA) is notably high, ranging from 36.3 to 43.0 per 100,000 population in females and from 87.7 to 135.4 per 100,000 population in males [4,5], significantly exceeding the rates reported in Western countries (<1.5 per 100,000 population) [6]. Despite recent advances in diagnostic imaging and surgical techniques, iCCA often remains asymptomatic until the disease reaches an advanced stage, leading to limited treatment options and poor clinical outcomes. Hepatic resection with curative intent remains the cornerstone of treatment; however, the long-term prognosis is dismal, with 5-year overall survival (OS) rates ranging from 10% to 35% [1,7,8]. A major challenge in iCCA management is the high rate of postoperative recurrence reported in up to 70% of patients with approximately 25% occurring within the first six months and nearly 50% within two years following surgery [9,10]. Early recurrence (ER), often defined as recurrence within 12–24 months after surgery, is associated with particularly poor outcomes and remains a major challenge in postoperative management [11,12,13].

Preoperative inflammatory markers such as neutrophil-to-lymphocyte ratio (NLR) and lymphocyte-to-monocyte ratio (LMR) reflect the host’s immune status and have shown prognostic relevance in hepatobiliary and other solid malignancies [14,15]. Likewise, immuno-nutritional indices including the prognostic nutritional index (PNI) and the Controlling Nutritional Status (CONUT) score have been explored for their role in predicting postoperative outcomes. The CONUT score, incorporating serum albumin, total cholesterol, and total lymphocyte count, provides a comprehensive measure of both nutritional and immune status [16,17]. Originally developed to screen for hospital malnutrition, it has since demonstrated independent prognostic value in gastrointestinal, pancreatic, hepatic, and even non-resectable CCA. In cancer research, particularly in iCCA, no universally accepted or clinically validated cut-off values exist for these inflammatory or immuno-nutritional indices. Reported thresholds vary substantially across studies, with NLR values in the approximate range of 1.93 to 5 frequently associated with adverse survival outcomes and an increased risk of recurrence, depending on the study population and analytical approach [14,15,18,19]. Likewise, although the CONUT score provides categorized ranges that may facilitate nutritional assessment, these scores have not been validated to contraindicate curative-intent resection in routine clinical practice. Accordingly, NLR, LMR, PNI, and CONUT should be regarded as adjunctive tools for perioperative risk stratification and postoperative management rather than determinants of surgical eligibility. Moreover, these indices may help identify patients who could benefit from prehabilitation or nutritional optimization before resection, potentially contributing to improved DFS and OS.

Despite the growing interest in preoperative inflammatory and immuno-nutritional biomarkers, robust comparative evidence remains limited regarding their relative prognostic performance for DFS and OS, particularly within clinically relevant subgroups. These include patients achieving margin-negative (R0) resection, those who subsequently develop recurrence, and patients with R0 resection who later experience recurrence. To date, no study has simultaneously compared multiple preoperative inflammatory and nutritional indices within the same population or systematically evaluated whether their prognostic utility persists across different disease-course contexts, including the post-recurrence setting.

Therefore, the objectives of this study were first to compare the prognostic associations of established preoperative inflammatory and immuno-nutritional indices, including NLR, LMR, PNI, and CONUT, with disease-free survival and overall survival in the overall cohort of patients with iCCA undergoing curative-intent hepatic resection. We subsequently evaluated the prognostic performance of these indices in clinically relevant subgroups, including patients achieving margin-negative (R0) resection, those who developed recurrence, and patients with R0 resection who subsequently experienced recurrence, in order to assess the consistency and context-specific utility of these commonly used preoperative scores across different surgical and disease-course settings. Given that these biomarkers are inexpensive, non-invasive, and routinely available in clinical practice, this approach aims to inform preoperative risk stratification and prognostic assessment in iCCA, while emphasizing that such indices should not be used as determinants of surgical eligibility.

## 2. Materials and Methods

### 2.1. Patients

This retrospective study included patients who underwent curative-intent resection for intrahepatic cholangiocarcinoma (iCCA) at Srinagarind Hospital, Khon Kaen University, Thailand, between January 2015 and December 2021. During the study period, a total of 220 patients with pathologically confirmed iCCA underwent curative-intent hepatic resection. Patients were selected for the present analysis according to predefined inclusion and exclusion criteria. Inclusion criteria were as follows: (i) curative-intent hepatic resection for iCCA, (ii) histopathological confirmation of iCCA, (iii) had an ECOG performance status of 0–1, (iv) did not receive neoadjuvant chemotherapy, and (v) had complete preoperative laboratory, clinicopathological, and follow-up data. Preoperative tumor biomarkers, including carcinoembryonic antigen (CEA) and carbohydrate antigen 19-9 (CA19-9), were collected when available and used for subsequent analyses. Exclusion criteria were as follows: (i) those with other biliary tract diseases prior to resection, (ii) patients who underwent repeat hepatectomy for recurrent disease, (iii) patients lost to follow-up after resection, and (iv) those with incomplete clinical or laboratory records. After applying these criteria, a total of 213 patients were eligible and included in the final analysis (Figure 1). The study protocol was approved by the Human Research Ethics Committee of Khon Kaen University (HE681665, approved on 3 November 2025) in accordance with the Declaration of Helsinki (1964) and its later amendments.

### 2.2. Recorded Data

All patients received a standardized preoperative oral nutritional supplement regimen prior to surgical intervention, based on previously established protocols. Blood samples were collected approximately one month before hepatic resection to assess baseline biochemical and hematological parameters, including liver function tests [aspartate aminotransferase (AST), alanine aminotransferase (ALT), and alkaline phosphatase (ALP)], total bilirubin, serum cholesterol, albumin, total white blood cell count, lymphocyte count, monocyte count, neutrophil count, and tumor markers [carcinoembryonic antigen (CEA) and carbohydrate antigen 19-9 (CA19-9)].

Intraoperative data collected included patient demographics (age and sex), the anatomical region of hepatic resection, specimen weight, tumor dimensions, gross tumor appearance, surgical margin status, and evidence of invasion into adjacent organs. Resected liver tissues were evaluated by experienced pathologists using standard protocols. Tissue samples were fixed in formalin and embedded in paraffin blocks. Sections with a thickness of 5 μm were prepared and stained with hematoxylin and eosin for microscopic examination.

Histopathological features were reviewed and classified in accordance with the 2019 World Health Organization (WHO) criteria for tumors of the digestive system [20]. Key histological parameters recorded included margin status, presence of lymphovascular invasion, lymph node involvement, and distant metastasis. All findings were cross-referenced with gross examination and used to determine pathological staging according to the 8th edition of the American Joint Committee on Cancer (AJCC)/International Union Against Cancer (UICC) staging system. Subsequently, patients were further categorized into four groups based on the tumor-node-metastasis (TNM) classification according to the 8th edition of the AJCC/UICC staging system [21]. Postoperative follow-up data were obtained either from Srinagarind Hospital or affiliated regional healthcare centers.

### 2.3. Tumor Morphology

Tumor morphology identification in iCCA was performed as described by Sa-ngiamwibool et al. [22]. Briefly, resected liver specimens were serially sectioned, photographed, and assessed for tumor growth patterns during gross examination. The patterns mass-forming (MF), periductal infiltrating (PI), and intraductal growth (ID) were recorded, and their proportions were estimated in 10% increments. Tumors were classified as having either a single growth pattern or a combination of patterns (e.g., ID + PI, ID + MF, PI + MF, or ID + PI + MF). The assigned growth patterns were subsequently confirmed by histopathological examination conducted by experienced pathologists.

### 2.4. Preoperative Score Assessment

Four preoperative inflammation- and nutrition-based prognostic scores were calculated using clinical and laboratory data obtained within one week prior to resection:▪Neutrophil-to-Lymphocyte Ratio (NLR): Calculated as the absolute neutrophil count divided by the absolute lymphocyte count [23].▪Lymphocyte-to-Monocyte Ratio (LMR): Calculated by dividing the absolute lymphocyte count by the absolute monocyte count [24].▪Prognostic Nutritional Index (PNI): Calculated using the following formula:PNI = [10 × serum albumin (g/dL)] + [0.005 × total lymphocyte count (/mm^3^)].

Lower PNI values reflect impaired nutritional and immunological status [25].

▪Controlling Nutritional Status (CONUT) Score: Derived from serum albumin level, total cholesterol concentration, and total lymphocyte count. The CONUT score was calculated according to previously established criteria, with higher scores indicating worse nutritional status [26].

All scores were analyzed both as continuous variables and as categorical variables, dichotomized into high and low groups based on previously validated cutoff values or the median values, as appropriate, whereas CONUT was classified based on established severity categories as described in our previous study [17].

### 2.5. Postoperative Follow-Up and Outcome Measurements

Following resection, patients underwent routine postoperative surveillance comprising physical examinations and imaging assessments, including computed tomography (CT) or magnetic resonance imaging (MRI) of the chest, abdomen, and pelvis, every three months during the first two years, every six months during years three to five, and annually thereafter, with additional evaluations performed if clinically indicated. The study endpoints included overall survival (OS), and disease-free survival (DFS). OS was calculated from the date of liver resection to either death or last follow-up. DFS was defined as the time from liver resection to the first documented recurrence of iCCA or death, whichever occurred first, or last follow-up, consistent with previous surgical oncology studies [27,28]. Recurrence was determined based on histopathological confirmation, radiologic evidence from ultrasound, CT, or MRI, or clinical documentation in the medical records.

### 2.6. Statistical Analysis

All statistical analyses were performed using SPSS version 27.0 (IBM Corp., Armonk, NY, USA). Continuous variables were presented as mean ± standard deviation (SD) for normally distributed data or median (interquartile range, IQR) for non-normally distributed data, and compared using the independent samples *t*-test or Mann–Whitney U test, as appropriate. Categorical variables were expressed as frequencies (percentages) and compared using the Chi-square test or Fisher’s exact test, depending on the expected counts. Survival analyses were conducted using the Kaplan–Meier method, and differences between groups were assessed with the log-rank test. Overall survival (OS) was defined as the time from resection to death or last follow-up, while disease-free survival (DFS) was defined as the time from resection to the first documented recurrence or last follow-up.

Univariate and multivariate analyses were performed using Cox proportional hazards regression models to identify independent prognostic factors. Variables with a *p*-value < 0.05 in univariate analysis were entered into the multivariate model. The hazard ratio (HR) and 95% confidence interval (CI) were calculated for each variable.

## 3. Results

### 3.1. Baseline Characteristics and Operative Variables

A total of 213 patients with pathologically confirmed intrahepatic cholangiocarcinoma (iCCA) who underwent curative-intent resection and had complete preoperative data were included in the final analysis, following appropriate exclusions based on tumor location and data availability (Figure 1).

Baseline clinical and laboratory characteristics are summarized in Table 1. The median age of the cohort was 64 years (range, 33–88). A total of 114 patients (53.5%) were aged ≥64 years, and 135 patients (63.4%) were male. The median body mass index (BMI) was 23.0 kg/m^2^ (range, 14.6–32.9). BMI ≥ 23 kg/m^2^ was observed in 105 patients (49.3%). Among the patients, 51.2% had serum cholesterol levels ≥ 189 mg/dL, and 67.1% had albumin levels ≥ 4.2 g/dL. Elevated total white blood cell count (≥7600 cells/mm^3^) was observed in 50.2% of patients, while 49.8% and 50.2% had elevated lymphocyte and neutrophil counts, respectively. Total monocyte count ≥ 520 cells/mm^3^ was found in 49.8%, and total bilirubin ≥ 0.4 mg/dL in 65.3% of patients. Elevated aspartate aminotransferase (AST, ≥27 U/L) and alanine aminotransferase (ALT, ≥23 U/L) were present in 54.5% and 53.5% of patients, respectively, while alkaline phosphatase (ALP) was ≥117 U/L in 50.7%. Regarding tumor markers, among those with available data, 50.6% had carcinoembryonic antigen (CEA) levels ≥ 4.36 ng/mL (*n* = 180), and 49.4% had carbohydrate antigen 19-9 (CA19-9) levels ≥ 20.90 U/mL (*n* = 174).

In terms of preoperative prognostic scores, patients were stratified using median values as cut-off points for NLR (≥2.4), LMR (≥3.6), and PNI (≥52), with approximately half of the cohort falling into the high-score groups: 50.2% for NLR, 51.2% for LMR, and 48.4% for PNI. For the CONUT score, severity-based classification was applied according to established guidelines, with 12.2% of patients categorized as having moderate to severe malnutrition (scores 5–12).

Postoperative pathological findings revealed that 41.8% of tumors had non-intraductal (non-ID) morphologies, including periductal infiltrating, mass-forming, and mixed types. Positive surgical margins (R1 status) were observed in 35.7% of patients. Moderate to poorly differentiated tumors were identified in 9.4%, lymph node metastasis (N1) in 27.7%, and advanced-stage disease (stage III–IV) in 47.4% of patients. Tumor recurrence was documented in 87 patients (40.8%).

### 3.2. Impact of Preoperative Prognostic Scores on DFS and OS of Intrahepatic Cholangiocarcinoma

To evaluate the prognostic value of preoperative inflammation- and nutrition-based scores in iCCA, univariate and multivariate Cox regression analyses were conducted for disease-free survival (DFS) and overall survival (OS). The assessed preoperative indices included the NLR, LMR, PNI, and CONUT score, analyzed alongside key postoperative clinicopathological factors (tumor morphology, tumor differentiation, surgical margin status, lymph node metastasis, and TNM staging).

Among these preoperative markers, NLR was the only independent predictor of both DFS and OS. Patients with NLR ≥ 2.4 had markedly shorter median DFS (18.1 months) and OS (25.4 months) than those with NLR < 2.4 (75.9 months and not reached, respectively). Elevated NLR remained independently associated with poorer DFS (HR = 1.66, 95% CI: 1.07–3.78, *p* = 0.025) and OS (HR = 1.94, 95% CI: 1.22–3.10, *p* = 0.006) (Table 2 and Table 3 and Figure 2). Although LMR and CONUT showed significance in univariate analysis, none retained significance in multivariate models.

Several postoperative tumor characteristics also showed strong prognostic relevance. Tumor morphology had one of the most pronounced impacts. Patients with mixed-type morphology without intraductal (ID) components showed the poorest outcomes (median DFS 14.9 months, OS 21.3 months). Mixed-type tumors with ID components also had inferior survival (median DFS 45.0 months, OS 39.7 months), whereas pure ID tumors did not reach median survival. In multivariate analysis, mixed-type without ID morphology independently predicted worse DFS (HR = 3.06, 95% CI: 1.68–5.58, *p* < 0.001) and OS (HR = 4.55, 95% CI: 2.20–9.42, *p* < 0.001), while mixed-type with ID morphology independently predicted poorer OS (HR = 2.82, 95% CI: 1.36–5.81, *p* = 0.005) (Table 2 and Table 3).

Surgical margin status was significantly associated with both outcomes. Patients with positive margins had shorter median DFS (11.7 months) and OS (18.2 months) compared with those with negative margins (86.5 and 90.5 months). Positive margins remained an independent predictor of DFS (HR = 2.33, 95% CI: 1.59–3.41, *p* < 0.001) and OS (HR = 1.70, 95% CI: 1.12–2.53, *p* = 0.009) (Table 2 and Table 3).

TNM stage and lymph node metastasis were significantly associated with DFS and OS in univariate analyses. Patients with advanced TNM stage (III–IV) had shorter DFS (14.9 months) and OS (20.6 months), although only OS remained significant in multivariate analysis (HR = 1.78, 95% CI: 1.06–2.99, *p* = 0.029). N1 also showed strong univariate associations but was not significant in multivariate models (Table 2 and Table 3).

Overall, these findings indicate that high preoperative NLR, together with adverse pathological features, non-ID tumor morphology, positive surgical margins, and advanced TNM stage, are independently associated with poorer prognosis following curative-intent resection for iCCA.

### 3.3. Impact of Preoperative Prognostic Scores on DFS and OS in Margin-Free Resected Intrahepatic Cholangiocarcinoma

To better understand the prognostic value of preoperative inflammation- and nutrition-based scores in iCCA, we focused our analysis on patients who underwent negative margin resection (R0). A total of 137 patients with R0 resection were included. By excluding cases with positive resection margins (R1), we aimed to reduce the confounding effect of residual tumor burden on survival outcomes. This approach allowed for a more accurate assessment of the impact of preoperative prognostic scores and pathological factors on DFS and OS in patients with curatively resected iCCA.

Among the preoperative markers, NLR remained the only independent predictor of both DFS and OS in margin-free resections. Patients with NLR ≥ 2.4 had significantly shorter median DFS (29.5 months) and OS (52.6 months) compared with those with NLR < 2.4 (DFS and OS not reached). In multivariate analysis, elevated NLR was independently associated with poorer DFS (HR = 1.66, 95% CI: 1.07–3.78, *p* = 0.004) and OS (HR = 2.11, 95% CI: 1.17–3.80, *p* = 0.014) (Table 4 and Table 5 and Figure 3). Other preoperative scores, including LMR, PNI, and CONUT, did not retain significance in multivariate models.

Postoperative tumor characteristics also showed strong prognostic relevance. Tumor morphology remained a key factor. Patients with mixed-type morphology without ID components had the poorest outcomes (median DFS 26.2 months, OS 29.5 months), while those with mixed-type tumors with ID components had intermediate survival (median DFS 47.8 months, OS 58.9 months), and pure ID tumors did not reach median survival. In multivariate analysis, mixed-type without ID morphology independently predicted worse DFS (HR = 3.32, 95% CI: 1.66–6.64, *p* < 0.001) and OS (HR = 4.83, 95% CI: 2.08–11.22, *p* < 0.001), while mixed-type with ID morphology independently predicted poorer OS (HR = 3.18, 95% CI: 1.09–4.58, *p* = 0.008) (Table 4 and Table 5).

Histological differentiation, lymph node metastasis, and TNM stage were significant in univariate analyses for both DFS and OS but did not remain independent predictors in multivariate models, although late-stage disease showed borderline significance for OS (HR = 2.01, 95% CI: 0.99–4.06, *p* = 0.052).

Thus, in patients with R0 resection, high preoperative NLR, together with adverse pathological features—particularly non-ID tumor morphology—remained independently associated with poorer DFS and OS, highlighting the prognostic utility of preoperative inflammatory status even in margin-negative resections (Table 4 and Table 5).

### 3.4. Preoperative Prognostic Score Association with Recurrence Patterns

To investigate prognostic factors influencing outcomes after recurrence, we conducted a subgroup analysis including only patients who experienced disease recurrence (*n* = 87). This approach allowed us to focus specifically on the recurrent disease setting and better understand the prognostic dynamics following recurrence.

For preoperative inflammatory and immunonutritional markers, NLR remained an independent predictor of post-recurrence survival. Patients with NLR ≥ 2.4 had shorter median OS (18.0 months) compared with those with NLR < 2.4 (28.7 months), and multivariate analysis confirmed its independent prognostic value (HR = 1.83, 95% CI: 1.10–3.05, *p* = 0.004) (Figure 4). LMR, PNI, and CONUT were not independently associated with OS after recurrence.

With respect to postoperative pathological factors, N1 and late-stage TNM (III–IV) were independently associated with poorer post-recurrence OS. Specifically, N1 patients had a median OS of 12.9 months versus 28.0 months for N0 (HR = 1.66, 95% CI: 1.04–2.64, *p* = 0.035), while late-stage disease had a median OS of 12.9 months versus 34.9 months for early-stage disease (HR = 1.83, 95% CI: 1.09–3.09, *p* = 0.002).

Other factors, including tumor morphology, surgical margin, histological differentiation, and CONUT, were significant in univariate analyses but did not remain independent predictors in multivariate analysis.

These results indicate that in recurrent iCCA, high preoperative NLR, along with N1 and advanced TNM stage, are key independent predictors of poor post-recurrence OS, highlighting the continued prognostic value of preoperative inflammatory status (Table 6).

### 3.5. Prognostic Impact of Preoperative Scores on OS in Margin-Free Resected Intrahepatic Cholangiocarcinoma with Recurrence

To specifically assess prognostic factors influencing OS in a more homogenous and clinically relevant subset, we conducted a subgroup analysis including 68 patients who underwent margin-free resection and subsequently developed recurrence. This approach aimed to eliminate the confounding effect of positive surgical margin and focus on preoperative factors impacting survival after recurrence in patients who underwent curative-intent resection.

In this subset, most preoperative inflammatory and immunonutritional markers (NLR, LMR, PNI) were not significantly associated with post-recurrence OS. Median OS was 24.9 months for NLR ≥ 2.4 versus 30.9 months for NLR < 2.4, but multivariate analysis did not show a significant independent effect. Similarly, LMR and PNI were not predictive of OS in this cohort. In contrast, CONUT score emerged as an independently associated with poor OS. Patients with moderate-severe CONUT had a markedly shorter median OS (3.2 months) compared with normal CONUT (28.0 months), and multivariate analysis confirmed its independent factor (HR = 4.01, 95% CI: 1.62–9.93, *p* = 0.003) (Table 7 and Figure 5).

Among postoperative pathological factors, mixed-type morphology without ID components was independently associated with worse OS (median 22.1 months; HR = 2.53, 95% CI: 1.01–6.32, *p* = 0.047). Other factors, including tumor differentiation, N1, and late-stage TNM, were not significant in multivariate analysis. These findings suggest that, in margin-free resections with recurrence, CONUT score and non-ID tumor morphology are the key independent prognostic factors for post-recurrence OS, whereas NLR, LMR, and PNI have limited predictive value in this context (Table 7).

These findings demonstrate that both CONUT score and tumor morphology are independently associated with post-recurrence outcomes, with impaired nutritional condition (as reflected by moderate–severe CONUT) and non-ID tumor types being significant prognostic factors for poor overall survival in margin-free resections with recurrence.

## 4. Discussion

This study represents one of the largest single-center analyses to evaluate the prognostic relevance of both a systemic inflammatory index (NLR) and a nutritional index (CONUT score) in patients undergoing curative-intent resection for intrahepatic cholangiocarcinoma (iCCA). Although associations between preoperative inflammatory indices and adverse prognosis in iCCA have been reported previously, the key novel contribution of the present study lies in the comparative assessment of these two distinct host-related factors across clinically meaningful post-surgical subgroups. We observed that elevated preoperative NLR was consistently associated with poorer disease-free survival (DFS) and overall survival (OS), and that this association persisted across important clinical subsets, including patients who achieved margin-negative (R0) resection and those who subsequently developed recurrence. In contrast, the prognostic association of the CONUT score appeared more selective and context dependent, with independent significance observed only in the subgroup of patients with R0 resection who experienced recurrence. Collectively, these findings indicate that preoperative inflammatory and nutritional indices are associated with long-term outcomes in resected iCCA, with NLR demonstrating more consistent prognostic relevance across patient subgroups.

In iCCA, preoperative scores reflecting inflammation and nutritional status provide practical tools for risk stratification and perioperative planning at the population level [14,15,16,17]. In this study, we evaluated NLR, LMR, PNI, and CONUT for their prognostic associations with disease-free survival (DFS) and overall survival (OS). In the absence of universally accepted clinical cut-off values for NLR, LMR, and PNI in iCCA, cohort-specific median values (2.4, 3.6, and 52, respectively) were applied to facilitate balanced group-level comparisons. In contrast, CONUT was categorized using standardized thresholds based on serum albumin, total lymphocyte count, and total cholesterol, classifying patients as normal (0–1), mild (2–4), or moderate/severe (≥5) [26]. These indices capture complementary aspects of host status: NLR and LMR reflect systemic inflammation and immune suppression [23,24]; PNI represents combined nutritional and immune status [25]; and CONUT provides an integrated assessment of overall nutritional condition [26]. As non-invasive and cost-effective measures, these indices support cohort-level preoperative risk stratification and inform perioperative planning, rather than individualized clinical decision-making. Subgroup analyses in patients achieving R0 resection and those who developed recurrence further clarified the context-specific prognostic relevance of these commonly used scores.

Within this framework of cohort-level risk stratification, our study demonstrates that an elevated preoperative neutrophil-to-lymphocyte ratio (NLR ≥ 2.4) is independently associated with poorer disease-free survival (DFS) and overall survival (OS) in patients undergoing curative-intent resection for iCCA. This association remained consistent in the overall cohort (DFS: HR = 1.66; OS: HR = 1.94) and persisted in clinically relevant subgroups, including patients achieving margin-negative (R0) resection (DFS: HR = 1.66; OS: HR = 2.11) and those who subsequently developed recurrence (OS: HR = 1.83). These findings highlight the prognostic relevance of NLR across the disease course of iCCA, even among patients with apparently favorable surgical outcomes. Compared with other inflammation- and nutrition-based indices evaluated in this study, including LMR, PNI, and CONUT, NLR demonstrated the most consistent association with survival outcomes. With regard to prognostic thresholds, the NLR cut-off value of 2.4 observed in this cohort lies within the range commonly reported in prior iCCA studies, where values between approximately 2–4 have been associated with adverse survival outcomes and increased recurrence risk [14,15,19]. These observations are further supported by a systematic review and meta-analysis including 15 studies (18 cohorts) with a total of 4123 cases with iCCA, which showed that elevated preoperative NLR using cut-off values ranging from 1.93 to 5 was significantly associated with shorter OS and RFS [18]. Similar associations between elevated NLR and adverse oncologic outcomes have also been reported across multiple solid malignancies, including colorectal cancer and hepatocellular carcinoma, with commonly used cut-off values generally ranging between 2 and 5 [29,30,31,32,33,34,35,36,37]. The overlap between these reported ranges and the threshold identified in the present iCCA cohort supports the biological plausibility of NLR as a marker reflecting unfavorable tumor–host interactions, rather than disease-specific mechanisms. Importantly, the identified NLR threshold should not be interpreted as an optimal or definitive cut-off for individual patient decision-making. Rather, it represents a cohort-level risk stratification point associated with differential survival outcomes. Clinically, NLR may help identify patients at relatively higher risk who could benefit from closer postoperative surveillance or consideration of adjuvant strategies when clinically appropriate. However, translation of these cohort-based associations into individualized patient management requires prospective validation, standardized thresholds, and integration with established clinicopathological and imaging-based factors.

These findings support the potential role of NLR as a prognostic biomarker reflecting aggressive tumor behavior and an unfavorable disease course in iCCA. NLR is widely regarded as an indicator of the balance between systemic inflammation and host immune response. An elevated NLR reflects a relative increase in circulating neutrophils and/or a reduction in lymphocytes, a profile that has been associated with tumor-promoting conditions and adverse clinical outcomes. Prior studies have shown that neutrophils can facilitate tumor progression through the secretion of pro-inflammatory cytokines, growth factors, and matrix-degrading enzymes, thereby promoting angiogenesis, invasion, and metastasis [38,39]. In addition, neutrophil-derived reactive oxygen species and arginase have been reported to suppress lymphocyte proliferation and cytotoxic activity, contributing to impaired anti-tumor immunity [40,41]. In contrast, lymphocytes, particularly cytotoxic T cells and natural killer cells, play a central role in tumor surveillance and immune-mediated tumor control. A reduced lymphocyte count, as reflected by a high NLR, has therefore been associated with compromised adaptive immunity and diminished tumor control [42]. Taken together, these biologically plausible mechanisms described in prior literature may help explain why elevated preoperative NLR has been consistently associated with poorer DFS and OS across multiple cancer types, including iCCA [18,30,36,43]. Notably, although patients achieving margin-negative (R0) resection are generally expected to have favorable outcomes, the occurrence of recurrence in a subset of these patients may reflect underlying aggressive tumor biology that is not fully captured by conventional pathological assessment [44]. It should be emphasized that the present study was observational in nature and did not directly investigate these biological mechanisms; therefore, the mechanistic interpretations discussed here are based on established evidence from prior studies rather than direct experimental validation within this cohort.

Importantly, we found that a higher preoperative CONUT score, especially in the moderate to severe range, was associated with poorer prognosis. This likely reflects underlying malnutrition and weakened physical condition prior to surgery, which could limit patients’ physiological reserves and impair immune responses [17,45,46,47]. Consequently, even after achieving R0 resection, patients with elevated CONUT scores who experienced recurrence demonstrated worse survival outcomes. In our study, the NLR was found to reflect systemic inflammation and immune status both key factors in tumor progression and metastasis. An elevated NLR indicates a pro-tumor inflammatory state and immunosuppression, which influence disease behavior across all stages of iCCA. This explains why NLR serves as a reliable prognostic marker for both DFS and OS, including in patients with margin-negative (R0) resections and those who develop recurrence. In contrast, the CONUT score reflects nutritional status and physiological reserve, incorporating serum albumin, total cholesterol, and lymphocyte count, which collectively indicate a patient’s ability to tolerate stress and maintain immune competence [26]. While systemic inflammation, as captured by NLR, is a key driver of tumor progression, adequate nutrition and physical robustness are essential for postoperative recovery, immune surveillance, and resistance to recurrent disease [43]. Among R0-resected patients who experience recurrence, a high CONUT score may reflect reduced capacity to cope with tumor burden and impaired anti-tumor immune response, factors that are not fully represented by inflammatory markers alone [17,48]. Therefore, while NLR broadly predicts outcomes related to tumor biology and systemic inflammation, CONUT provides complementary prognostic information, particularly in recurrent cases, where the interplay between nutritional status, physiological reserve, and tumor aggressiveness critically influences survival. This distinction emphasizes the complementary role of NLR and CONUT in risk stratification and supports their potential use in guiding tailored postoperative management strategies in iCCA. It is important to emphasize that the decision to perform curative-intent hepatic resection for iCCA is fundamentally based on multidimensional clinical evaluation rather than any single biomarker. Operability depends on imaging-based resectability including vascular involvement, biliary extension, and the feasibility of achieving an adequate future liver remnant along with hepatic functional reserve, physiological fitness, performance status, and comorbidities. Consequently, inflammatory and immuno-nutritional biomarkers such as NLR and CONUT should be regarded as adjunctive prognostic tools rather than determinants of surgical eligibility. To date, no validated biomarker thresholds exist that would contraindicate curative-intent resection in iCCA or other solid tumors. These indices therefore support perioperative risk stratification but should not be used as stand-alone criteria to deny or permit resection.

Although these biomarkers should not determine surgical eligibility, their prognostic information has practical implications for perioperative management. Patients presenting with markedly elevated NLR or impaired nutritional indices may be considered a higher-risk subgroup who could benefit from closer postoperative surveillance, more intensive follow-up imaging, or prioritization for adjuvant systemic therapy when appropriate. Similarly, identifying poor nutritional reserve through CONUT may help guide targeted nutritional counseling or optimization before major hepatectomy, even though such measures are unlikely to reverse tumor-driven inflammation. In this context, structured prehabilitation strategies incorporating nutritional optimization and physical conditioning may represent a potentially beneficial future application to enhance perioperative resilience and postoperative recovery. However, such interventions were not systematically assessed in the present retrospective cohort and should be evaluated in future prospective studies. Thus, the clinical value of these indices lies in risk stratification and individualized perioperative planning rather than modifying or delaying resection based on biomarker normalization.

A major strength of this study is its practical, clinically oriented evaluation of preoperative inflammatory and immuno-nutritional indices for cohort-level risk stratification in patients undergoing curative-intent resection for intrahepatic cholangiocarcinoma. The purpose of this work was not to guide surgical eligibility or individualized treatment decisions, but to clarify how routinely available laboratory markers relate to long-term outcomes in real-world clinical practice. By analyzing these indices in the overall cohort and in clinically relevant subgroups, including patients with margin-negative (R0) resection and those who later developed recurrence, we assessed their prognostic value while accounting for major clinical factors that influence survival. Our findings show that elevated preoperative NLR is consistently associated with poorer survival across subgroups, suggesting that it reflects a general host-related risk linked to systemic inflammation. In contrast, the prognostic impact of the CONUT score was more context dependent, indicating that nutritional status provides additional prognostic information in selected clinical situations, particularly after recurrence. Because these indices are based on simple, low-cost laboratory tests, they may support perioperative risk stratification and postoperative surveillance planning, rather than serving as criteria for surgical decision-making.

Nevertheless, several limitations should be acknowledged. First, the retrospective, single-center design may introduce selection bias and limit the generalizability of the findings. Second, the relatively small sample size, particularly in subgroup analyses, may reduce statistical power and increase the risk of type II error. Third, several clinically relevant preoperative variables, including imaging-based tumor characteristics (such as tumor size and macroscopic morphology) and serum tumor biomarkers, were incompletely recorded, precluding their inclusion in the multivariable Cox regression models. This limitation may have restricted evaluation of the independent prognostic contribution of tumor-related factors and may partly explain the predominant significance observed for inflammatory and nutritional indices. In addition, standardized data on preoperative prehabilitation were not available, preventing assessment of its potential effects on immune-nutritional status and clinical outcomes. Fourth, information on perioperative and adjuvant treatments, including adjuvant chemotherapy, was not systematically documented during the earlier study period, limiting the ability to account for their potential impact on survival outcomes. Similarly, owing to the retrospective design and interval-based routine surveillance, some imprecision in determining the exact timing of recurrence was unavoidable, particularly during long-term follow-up. Fifth, although inflammatory and immunonutritional indices were dichotomized using cohort-specific cut-off values to facilitate clinical interpretability and group-level risk stratification, this approach may have reduced granularity and limited individualized prognostic assessment. Sixth, longitudinal postoperative measurements of inflammatory and nutritional indices were not available; therefore, it was not possible to determine whether preoperative abnormalities in NLR, LMR, PNI, and CONUT normalized after tumor surgery or persisted, limiting interpretation regarding tumor-driven versus host-related effects. Finally, post-recurrence treatments, including chemotherapy, radiotherapy, and redo surgery, were incompletely documented and could not be reliably categorized or analyzed, and thus were not included in the present study.

Future prospective multicenter studies with standardized data collection are warranted to validate these findings and refine preoperative prognostic models in iCCA. Integration of comprehensive preoperative imaging, serum tumor biomarkers, inflammatory and nutritional indices, continuous biomarker modeling, standardized perioperative management including prehabilitation, as well as post-recurrence treatment data and longitudinal postoperative assessments may enable more accurate recurrence determination, clarify tumor- versus host-driven effects, and improve individualized risk stratification and postoperative management.

## 5. Conclusions

This study demonstrates that elevated preoperative NLR is an independent and consistent factor associated with poorer DFS and OS in patients with iCCA undergoing curative-intent resection. The prognostic association of NLR persisted across clinically relevant subgroups, including patients who achieved margin-negative resection and those who subsequently developed recurrence, supporting its value as a broadly applicable prognostic marker reflecting adverse tumor biology and systemic inflammatory status. In contrast, the association of the CONUT score with outcome was limited and context dependent, with independent significance observed only in patients with margin-negative resection who later experienced recurrence, suggesting a selective prognostic association of nutritional status in specific clinical settings. Taken together, these findings indicate that preoperative systemic inflammation is strongly associated with long-term prognosis, whereas nutritional impairment may exert additional prognostic influence only in selected patient subsets. Accordingly, NLR may be useful for general preoperative risk stratification, while the CONUT score may provide supplementary prognostic information in specific clinical contexts. However, as an observational study, these findings demonstrate correlation, and further mechanistic studies are warranted to explore whether targeted interventions can influence these outcomes.

## Figures and Tables

**Figure 1 medsci-14-00023-f001:**
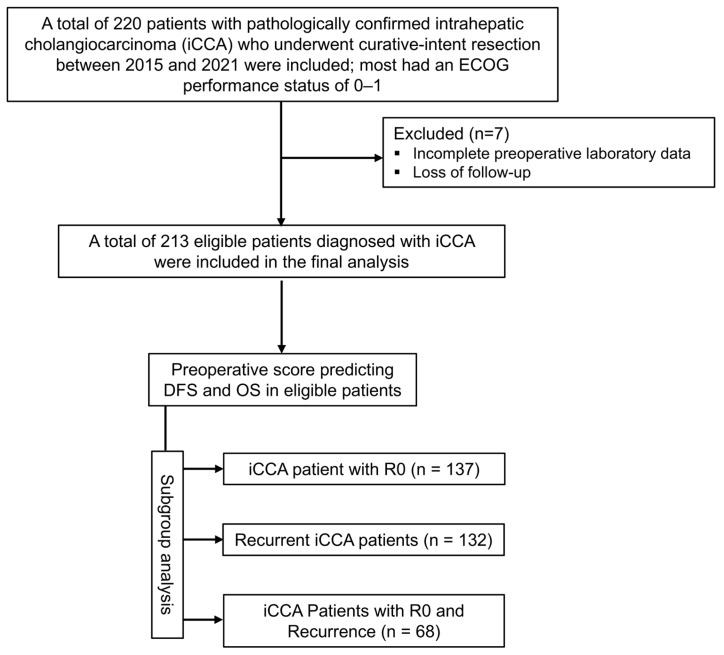
CONSORT flow diagram depicting the preoperative score to predict outcomes of intrahepatic cholangiocarcinoma.

**Figure 2 medsci-14-00023-f002:**
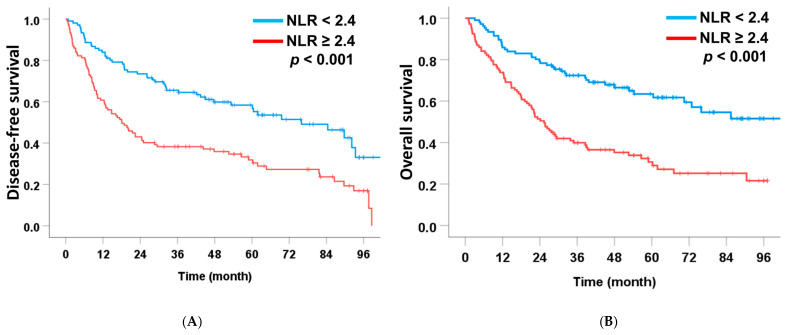
Association between significant preoperative scoring system and disease-free survival (**A**) and overall survival (**B**) in patients with intrahepatic cholangiocarcinoma undergoing curative-intent liver resection.

**Figure 3 medsci-14-00023-f003:**
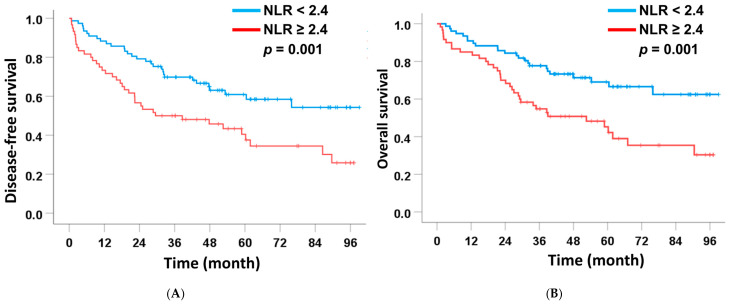
Association between significant preoperative scoring systems and disease-free survival (**A**) and overall survival (**B**) in the subgroup of patients with intrahepatic cholangiocarcinoma who underwent curative-intent liver resection with negative surgical margins.

**Figure 4 medsci-14-00023-f004:**
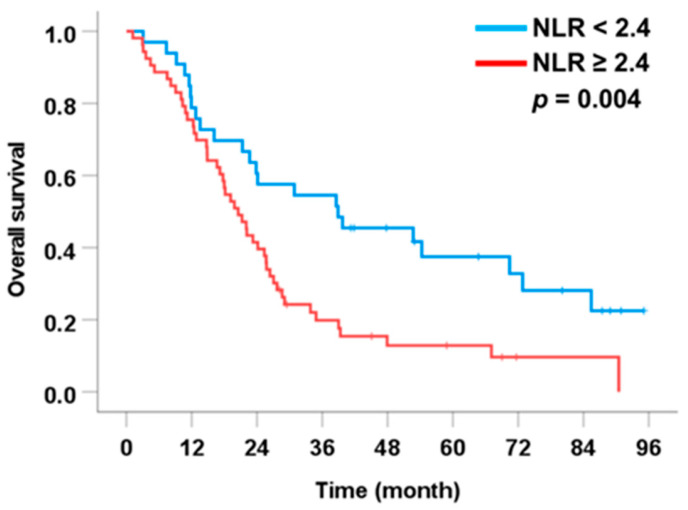
Association between significant preoperative scoring systems and overall survival in patients with intrahepatic cholangiocarcinoma who developed tumor recurrence after curative liver resection.

**Figure 5 medsci-14-00023-f005:**
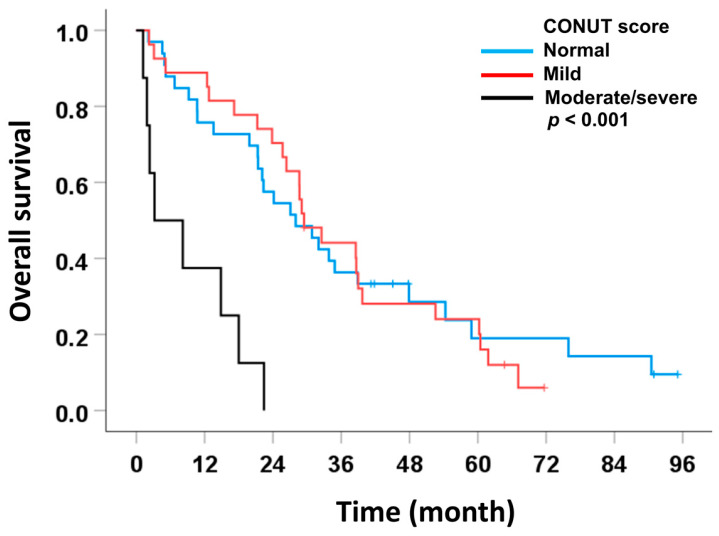
Association between significant preoperative scoring systems and overall survival in patients with intrahepatic cholangiocarcinoma who underwent curative-intent liver resection with negative surgical margins and subsequently developed recurrence.

**Table 1 medsci-14-00023-t001:** Clinicopathological characteristics.

Variable	No. of Patients (*n* = 213), *n* (%)
Age, median (range), years: 64 (33–88) Age ≥ 64 years	114 (53.5)
Gender, male	135 (63.4)
BMI, median (range), kg/m^2^: 23 (14.57–32.89) BMI ≥ 23 kg/m^2^	105 (49.3)
Preoperative factor, median (min-max)	
Cholesterol, ≥189 mg/dL	109 (51.2)
Albumin, ≥4.2 g/dL	143 (67.1)
Total WBC, ≥7600 cells/mm^3^	107 (50.2)
Total lymphocytes, ≥1927 cells/mm^3^	106 (49.8)
Total neutrophils, ≥4558 cells/mm^3^	107 (50.2)
Total monocyte count, ≥520 cells/mm^3^	106 (49.8)
Total bilirubin, ≥0.4 mg/dL	139 (65.3)
AST, ≥27 U/L	116 (54.5)
ALT, ≥23 U/L	114 (53.5)
ALP, ≥117 U/L	108 (50.7)
CEA ^#^ (*n* = 180), ≥4.36 ng/mL	91 (50.6)
CA19-9 ^#^ (*n* = 174), ≥20.90 U/mL	86 (49.4)
Preoperative score	
NLR, ≥2.4	107 (50.2)
LMR, ≥3.6	109 (51.2)
PNI, ≥52	103 (48.4)
CONUT, moderate/severe (5–8/9–12)	26 (12.2)
Postoperative factor	
Tumor morphology, without ID (PI, MF and Mix)	89 (41.8)
Surgical margin, R1	76 (35.7)
Histological differentiation, moderate/poorly	20 (9.4)
Lymph node metastasis, N1	59 (27.7)
TNM staging, Late stage (III–IV)	101 (47.4)
Recurrent patient	132 (62)

BMI: body mass index; WBC: white blood cell count; AST: aspartate aminotransferase; ALT: alanine aminotransferase; ALP: alkaline phosphatase; CEA: carcinoembryonic antigen; CA19-9: carbohydrate antigen 19-9; NLR: neutrophil-to-lymphocyte ratio; LMR: lymphocyte-to-monocyte ratio; PNI: prognostic nutritional index; CONUT: controlling nutritional status score; ID: intraductal growth; PI: periductal infiltrating; MF: mass-forming; TNM: tumor–node–metastasis classification. ^#^ CEA and CA19-9 were not measured in some patients.

**Table 2 medsci-14-00023-t002:** Univariate and multivariate analyses for disease-free survival in intrahepatic cholangiocarcinoma patients.

Variable	Median DFS (Month)	Univariate	Multivariate	
		*p*-Value	HR (95% CI)	*p*-Value
Tumor morphology				
ID	NR		1	
Mix type with ID	45.0	<0.001	1.73 (0.94–6.24)	0.077
Mix type without ID	14.9	<0.001	3.06 (1.68–5.58)	<0.001
Surgical Margin				
Negative	86.5		1	
Positive	11.7	<0.001	2.33 (1.59–3.41)	<0.001
Histological differentiation				
Well	52.6		1	
Moderate/poorly	14.8	0.035	1.11 (0.62–1.98)	0.729
Lymph node metastasis (N)				
N0	60.2		1	
N1	13.5	<0.001	1.12 (0.68–1.82)	0.647
TNM Stage				
Early stage (I–II)	84.4		1	
Late stage (III–IV)	14.9	<0.001	1.47 (0.90–2.42)	0.125
Preoperative score				
NLR				
<2.4	75.9		1	
≥2.4	18.1	<0.001	1.66 (1.07–3.78)	0.025
LMR				
<3.6	20.3		1	
≥3.6	61.8	0.012	1.06 (0.678–1.66)	0.798
PNI				
<52	32.3		-	
≥52	60.2	0.365	-	-
CONUT				
Normal	61.8		1	
Mild	42.2	0.296	-	-
Moderate-Severe	8.4	0.005	1.04 (0.58–1.89)	0.889

ID: intraductal growth; NLR: neutrophil-to-lymphocyte ratio; LMR: lymphocyte-to-monocyte ratio; PNI: prognostic nutritional index; CONUT: controlling nutritional status score; TNM: tumor–node–metastasis classification. NR = Not reached; median disease-free survival could not be estimated because the cumulative disease-free survival did not drop below 0.5.

**Table 3 medsci-14-00023-t003:** Univariate and multivariate analyses for overall survival in intrahepatic cholangiocarcinoma patients.

Variable	Median Survival (OS; Month)	Univariate	Multivariate	
		*p*-Value	HR (95% CI)	*p*-Value
Tumor morphology				
ID	NR		1	
Mix type with ID	39.7	<0.001	2.82 (1.36–5.81)	0.005
Mix type without ID	21.3	<0.001	4.55 (2.20–9.42)	<0.001
Surgical Margin				
Negative	90.5		1	
Positive	18.2	<0.001	1.70 (1.12–2.53)	0.009
Histological differentiation				
Well	52.7		-	
Moderate/poorly	29.5	0.519	-	-
Lymph node metastasis (N)				
N0	75.9		1	
N1	15.9	<0.001	1.53 (0.93–2.51)	0.094
TNM Stage				
Early stage (I–II)	NR		1	
Late stage (III–IV)	20.6	<0.001	1.78 (1.06–2.99)	0.029
Preoperative score				
NLR				
<2.4	NR		1	
≥2.4	25.4	<0.001	1.94 (1.22–3.10)	0.006
LMR				
<3.6	27.1		1	
≥3.6	85.4	<0.001	0.81 (0.51–1.29)	0.373
PNI				
<52	38.9		-	
≥52	58.9	0.242	-	-
CONUT				
Normal	75.9		1	
Mild	52.7	0.432	-	-
Moderate-Severe	12.3	0.001	0.81 (0.43–1.51)	0.505

ID, intraductal growth; NLR, neutrophil-to-lymphocyte ratio; LMR, lymphocyte-to-monocyte ratio; PNI, prognostic nutritional index; CONUT, controlling nutritional status score; TNM, tumor–node–metastasis classification. NR = Not reached; median survival could not be estimated because the cumulative survival did not drop below 0.5.

**Table 4 medsci-14-00023-t004:** Univariate and multivariate analyses for DFS in margin-free resected intrahepatic cholangiocarcinoma.

Variable	Median DFS (Month)	Univariate	Multivariate	
		*p*-Value	HR (95% CI)	*p*-Value
Tumor morphology				
ID	NR		1	
Mix type with ID	47.8	0.011	2.23 (1.09–4.58)	0.029
Mix type without ID	26.2	<0.001	3.32 (1.66–6.64)	<0.001
Histological differentiation				
Well	75.9		1	
Moderate/poorly	29.5	0.036	1.35 (0.60–3.06)	0.473
Lymph node metastasis (N)				
N0	86.5		1	
N1	22.4	<0.001	1.61 (0.81–3.20)	0.172
TNM Stage				
Early stage (I–II)	90		1	
Late stage (III–IV)	23.1	<0.001	1.36 (0.71–2.60)	0.351
Preoperative score				
NLR				
<2.4	NR		1	
≥2.4	29.5	0.003	1.66 (1.07–3.78)	0.004
LMR				
<3.6	47.8		-	
≥3.6	NR	0.052	-	-
PNI				
<52	53.3		-	
≥52	75.9	0.530	-	-
CONUT				
Normal	58.9		-	
Mild	61.8	0.833	-	-
Moderate-Severe	12.4	0.162	-	-

ID, intraductal growth; NLR, neutrophil-to-lymphocyte ratio; LMR, lymphocyte-to-monocyte ratio; PNI, prognostic nutritional index; CONUT, controlling nutritional status score; TNM, tumor–node–metastasis classification. NR = Not reached; median survival could not be estimated because the cumulative survival did not drop below 0.5.

**Table 5 medsci-14-00023-t005:** Univariate and multivariate analyses for OS in margin-free resected intrahepatic cholangiocarcinoma.

Variable	Median Survival (OS; Month)	Univariate	Multivariate	
		*p*-Value	HR (95% CI)	*p*-Value
Tumor morphology				
ID	NR		1	
Mix type with ID	58.9	<0.001	3.18 (7.45)	0.008
Mix type without ID	29.5	<0.001	4.83 (2.08–11.22)	<0.001
Histological differentiation				
Well	90.5		1	
Moderate/poorly	29.5	0.037	1.55 (0.63–3.84)	0.340
Lymph node metastasis (N)				
N0	NR		1	
N1	23.9	<0.001	1.43 (0.68–3.01)	0.346
TNM Stage				
Early stage (I–II)	NR		1	
Late stage (III–IV)	29.5	<0.001	2.01 (0.99–4.06)	0.052
Preoperative score				
NLR				
<2.4	NR		1	
≥2.4	52.6	0.001	2.11 (1.17–3.80)	0.014
LMR				
<3.6	52.6		1	
≥3.6	NR	0.008	0.64 (0.36–1.16)	0.139
PNI				
<52	NR		-	
≥52	75.9	0.871	-	-
CONUT				
Normal	90.5		1	
Mild	67.1	0.922	-	-
Moderate-Severe	18	0.030	1.43 (0.60–3.39)	0.422

ID, intraductal growth; NLR, neutrophil-to-lymphocyte ratio; LMR, lymphocyte-to-monocyte ratio; PNI, prognostic nutritional index; CONUT, controlling nutritional status score; TNM, tumor–node–metastasis classification. NR = Not reached; median survival could not be estimated because the cumulative survival did not drop below 0.5. NR = Not reached; median survival could not be estimated because the cumulative survival did not drop below 0.5.

**Table 6 medsci-14-00023-t006:** Univariate and multivariate analyses for OS in intrahepatic cholangiocarcinoma who developed tumor recurrence after curative liver resection.

Variable	Median Survival (OS; Month)	Univariate	Multivariate	
		*p*-Value	HR (95% CI)	*p*-Value
Tumor morphology				
ID	60.5			
Mix type with ID	25.4	0.004	1.59 (0.74–3.41)	0.234
Mix type without ID	14.7	0.001	1.77 (0.81–3.85)	0.151
Surgical Margin				
Negative	27.1		-	
Positive	17.4	0.088	-	-
Histological differentiation				
Well	21.5		-	
Moderate/poorly	21.3	0.792	-	-
Lymph node metastasis (N)				
N0	28.0		1	
N1	12.9	<0.001	1.66 (1.04–2.64)	0.035
TNM Stage				
Early stage (I–II)	34.9		1	
Late stage (III–IV)	12.9	<0.001	1.83 (1.09–3.09)	0.002
Preoperative score				
NLR				
<2.4	28.7		1	
≥2.4	18.0	0.002	1.83 (1.10–3.05)	0.004
LMR				
<3.6	18.0		1	
≥3.6	25.4	0.014	0.77 (0.37–1.62)	0.494
PNI				
<52	18.2		-	
≥52	24.1	0.513	-	-
CONUT				
Normal	22.3		1	
Mild	25.7	0.313	-	
Moderate-Severe	8.2	0.002	1.81 (0.63–5.12)	0.269

ID, intraductal growth; NLR, neutrophil-to-lymphocyte ratio; LMR, lymphocyte-to-monocyte ratio; PNI, prognostic nutritional index; CONUT, controlling nutritional status score; TNM, tumor–node–metastasis classification. NR = Not reached; median survival could not be estimated because the cumulative survival did not drop below 0.5. NR = Not reached; median survival could not be estimated because the cumulative survival did not drop below 0.5.

**Table 7 medsci-14-00023-t007:** Preoperative scores associated with overall survival in iCCA patients with recurrence and margin-negative curative-intent resection.

Variable	Median Survival (OS; Month)	Univariate	Multivariate	
		*p*-Value	HR (95% CI)	*p*-Value
Tumor morphology				
ID	60.4		1	
Mix type with ID	26.4	0.009	2.06 (0.82–5.20)	0.124
Mix type without ID	22.1	0.002	2.53 (1.01–6.32)	0.047
Histological differentiation				
Well	28.0		-	
Moderate/poorly	12.4	0.348	-	-
Lymph node metastasis (N)				
N0	29.1		1	
N1	21.3	0.003	1.10 (0.53–2.25)	0.802
TNM Stage				
Early stage (I–II)	34.9		1	
Late stage (III–IV)	21.3	<0.001	1.72 (0.80–3.70)	0.168
Preoperative score				
NLR				
<2.4	30.9		-	
≥2.4	24.9	0.163	-	-
LMR				
<3.6	26.4		-	
≥3.6	34.9	0.314	-	-
PNI				
<52	25.7		-	
≥52	28.6	0.737	-	-
CONUT				
Normal	28.0		1	
Mild	29.5	0.386	-	-
Moderate-Severe	3.2	<0.001	4.01 (1.62–9.93)	0.003

ID, intraductal growth; NLR, neutrophil-to-lymphocyte ratio; LMR, lymphocyte-to-monocyte ratio; PNI, prognostic nutritional index; CONUT, controlling nutritional status score; TNM, tumor–node–metastasis classification.

## Data Availability

The original contributions presented in this study are included in the article. Further inquiries can be directed to the corresponding author(s).

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
