# Peer review of "Preoperative Prognostic Score for Patients with Intrahepatic Cholangiocarcinoma Undergoing Curative-Intent Resection"

_medsci, 2026, doi:10.3390/medsci14010023_

Round 1
Reviewer 1 Report
Comments and Suggestions for Authors
Preoperative Prognostic Score for Patients with Intrahepatic Cholangiocarcinoma Undergoing Curative Resection
Jarin Chindaprasirt et al
Abstract
- Review of patients undergoing “curative resection”. Were those with R1 resection or intra-op resection of nodal metastases removed from analysis?
- The authors suggest The CONUT score also independently predicted OS in both R0 and recurrent subgroups. I suspect it correlated but was not shown to predict
- An important distinction. Prediction would require a level (a threshold) which reproducibly was associated with better or worse outcome.
- The histological correlate and early recurrence correlation with survival should be identified as being similar to prior publications.
- The conclusion implies that the systemic inflammatory response is related to nutrition but the abstract does not provide data to support this interpretation.
Introduction
Absence of key issues;
- The novelty of this study. What exactly has not previously been analysed?
- Can prehabilitation reverse the immune and nutritional profiles and improve outcome.?
- Have any thresholds been identified to suggest levels of biomarkers that should / would contraindicate resection of iCCA or other cancers?
Methods
- If the study is specific to iCCA why was the wider cohort with pCCA and dCCA mentioned. They should be excluded or the reason for inclusion identified.
- Preoperative nutrition was given. The duration and reason should be identified and how the calories given correlated with nutritional status.
- Pre-op modulation? eg biliary drainage, residual liver measurement, PVE/HVE utilized.
- Was there a specialist MDT.? What was the pre-op radiology TNM stage for the cohort.
- What was the purpose of the growth pattern evaluation by a non pathologist?
- Unusual to label all deaths to be due to cancer recurrence, even when no radiological evidence of recurrence. Defining the nature of the terminal event could lead to a more accurate assessment of recurrence.
Results
- Should be labelled as potentially curative resection as the majority recur.
- Median age of cohort confusing as appears to be applied to a sub group.
- BMI should be given as median and range
- NLR, LMR, PNI and CONUT have been grouped as low or high based on median values. This produces only two cohorts and does not allow useful prognostic information for individual patients which would need continuous variables to be correlated with outcome.
- No previously validated cut off values for theprognostic markers have been applied.
- The NLR high vs low values show a striking association with survival and the data excluding confounders such as involved margins and outcome of those with recurrence of interest.
- Would be useful to analyse those with recurrence both from baseline and from diagnosis of recurrence.
- Conclusion to results should be that a correlation has been shown with outcome in resected iCCA but there is no evidence that it determined outcome as suggested by authors. This would require a mechanistic rather than observational study.
- Notable absences from data analysis
- Effect of adjuvant chemotherapy and /or immunotherapy on cohort survival.?
- Did prognostic scores normalize following recovery from the iCCA resection ie were the parameters NLR, LMR, PNI, CONUT secondary to the iCCA per se.
Discussion
Para 1 : Authors should identify the novel findings of their study in comparison to prior reports.
Para 2 : This is background and not findings from this study, Should be moved to introduction or removed.
Para 3: Authors suggest immune/nutritional parameters used in protocol for risk stratification and pre-op planning. Need to justify the use of median values to form two groups for comparison and to discuss the major limitations of this approach for prognostic assessment of an individual. How the grouped data would be applied is unclear.
Para 4: Discussion should focus on how the indices could be applied to iCCA rather than discussion of other disease groups where the prognostic indices correlate with survival.
Para 5. Optimal scores have not been determined as suggested, only cancer risks associated with low or high scores. How would cohort data be translated into some clinical value for individual patient management.
Para 6 : Mechanistic discussion important but current study has not contributed to our understanding of mechanism. CONUT discussion important but authors should clarify the numbers in the moderate/ severe group in whom survival is reduced. Again no thresholds identified which could be used in clinical practise.
Para 7 : Some insight into limitation of study but no clarity on how these findings will be applied to patients selection for surgery, prehabilitation, nutrition supplementation or post op management.
Figures: Helfpful and clear
Tables: Require explanation of abbreviations in foot notes.
References : Relevant and up to date

Author Response
Overall response and summary of revisions
We sincerely thank the Editor and all reviewers for their constructive and insightful comments, which improved the clarity, methodological transparency, and clinical interpretation of our manuscript entitled “Preoperative Prognostic Score for Patients with Intrahepatic Cholangiocarcinoma Undergoing Curative-Intent Resection.” We have addressed all points and revised the manuscript accordingly. The major revisions include: (1) clarification of cohort definition and surgical intent, (2) correction of terminology and internal inconsistencies, (3) strengthening of the Methods and endpoint definitions, (4) improved presentation and consistency of Results and tables, and (5) refinement of the Discussion/Conclusion to avoid causal language and emphasize cohort-level risk stratification and appropriate clinical applicability. Detailed point-by-point responses are provided below.
Key revisions across the manuscript
- Terminology and cohort definition: Replaced “curative resection” with “curative-intent resection” throughout to reflect intent at resection (with margin and nodal status confirmed postoperatively). Clarified inclusion of R1 resections and resected nodal metastases when no gross residual disease remained, ensuring consistency across Abstract/Methods/Results.
- Methods and outcomes: Clarified that only iCCA cases were included and updated Figure 1 to show exclusions of pCCA/dCCA. Revised DFS definition to standard practice (time to recurrence or death from any cause) and removed the assumption that deaths without recurrence were cancer-related. Clarified recurrence ascertainment using routine surveillance and acknowledged that recurrence timing reflects first detection.
- Data consistency and corrections: Removed erroneous references to PLR (not analyzed), corrected “NLP” to “NLR,” and revised statements (univariate significance for PNI) to match the tables.
- Results presentation: Clarified reporting of age to avoid subgroup misinterpretation, revised BMI reporting (median and range), and added complete abbreviation definitions in table footnotes.
- Cut-offs and limitations: Justified the use of median-based cut-offs for NLR/LMR/PNI in the absence of validated iCCA thresholds, and explicitly noted that dichotomization supports cohort comparisons but limits individualized prognostic inference (with continuous-variable analyses addressed where applicable).
- Clinical interpretation: Revised Discussion/Conclusion to emphasize association (not causation) and that mechanistic confirmation requires dedicated studies. Clarified that CONUT findings were context-dependent (mainly in the R0 + recurrence subgroup) and that indices should be used as adjunctive tools for cohort-level risk stratification, not determinants of surgical eligibility.
- Limitations and future directions: Acknowledged missing/incomplete data (imaging tumor features, tumor markers, adjuvant/post-recurrence treatment, longitudinal indices) and highlighted the need for prospective studies with standardized data capture and evaluation of prehabilitation/nutrition protocols.
Reviewer 1
Open Review
() I would not like to sign my review report
(x) I would like to sign my review report
Quality of English Language
( ) The English could be improved to more clearly express the research.
(x) The English is fine and does not require any improvement.
|
Yes |
Can be improved |
Must be improved |
Not applicable |
|
|
Does the introduction provide sufficient background and include all relevant references? |
( ) |
(x) |
( ) |
( ) |
|
Is the research design appropriate? |
( ) |
(x) |
( ) |
( ) |
|
Are the methods adequately described? |
( ) |
( ) |
(x) |
( ) |
|
Are the results clearly presented? |
( ) |
(x) |
( ) |
( ) |
|
Are the conclusions supported by the results? |
( ) |
( ) |
(x) |
( ) |
|
Are all figures and tables clear and well-presented? |
(x) |
( ) |
( ) |
( ) |
Comments and Suggestions for Authors
Preoperative Prognostic Score for Patients with Intrahepatic Cholangiocarcinoma Undergoing Curative Resection
Jarin Chindaprasirt et al
Abstract
- Review of patients undergoing “curative resection”. Were those with R1 resection or intra-op resection of nodal metastases removed from analysis?
Author response Q1: We thank the reviewer for this important clarification. In the present study, all patients underwent resection with curative intent, based on preoperative imaging and intraoperative assessment indicating that complete macroscopic tumor removal was achievable. Importantly, margin status (R0 vs. R1) and lymph node status (N0 vs. N1) can only be definitively determined by final pathological examination and therefore cannot be known with certainty at the time of resection.
Accordingly, patients with microscopic margin positivity (R1), identified only on final pathology, were included, as margin status does not alter the surgical intent. Similarly, patients with lymph node metastases identified intraoperatively or on pathological examination were also included, provided that metastatic nodes were completely resected and no gross residual disease (R2 resection) remained.
To improve clarity and more accurately reflect the surgical decision-making process, we have revised the terminology throughout the manuscript from “curative resection” to “curative-intent resection.”
- The authors suggest The CONUT score also independently predicted OS in both R0 and recurrent subgroups. I suspect it correlatedbut was not shown to predict
Author response Q2: We appreciate the reviewer’s comment. We have revised the wording accordingly and now state that the CONUT score correlated with OS rather than independently predicted OS in the R0 and recurrent subgroups.
- An important distinction. Prediction would require a level (a threshold) which reproducibly was associated with better or worse outcome.
Author response Q3: Thank you for highlighting this important distinction regarding prediction and the requirement for reproducible thresholds. With respect to the NLR, several prior studies in iCCA and other solid tumors have identified prognostic cut-off values typically ranging between 2 and 4, and these thresholds have demonstrated reproducible associations with survival outcomes. The median-based cut-off used in our cohort (NLR = 2.4) falls within this well-established range and is consistent with values applied in multiple previous publications, supporting its validity for prognostic stratification in this setting.
For the CONUT score, we applied the standardized grading system (normal, mild, moderate/severe) that has been used across prior studies evaluating nutritional and immunologic status. However, because the number of patients falling into the moderate/severe CONUT category was relatively small, especially in subgroup analyses, the score did not consistently retain independent predictive value. In accordance with the reviewer’s suggestion, we have revised the manuscript to describe CONUT as correlated with overall survival rather than predictive, and we avoid implying any reproducible prognostic threshold.
- The histological correlate and early recurrence correlation with survival should be identified as being similar to prior publications.
Author Respond Q4: Thank you very much for this helpful comment. In an earlier version of our internal analysis, we explored survival after recurrence by subdividing patients who experienced recurrence into early and late recurrence using a 365-day cutoff. However, after methodological review, we intentionally removed this analysis from the Results for both statistical and clinical reasons. First, timing of recurrence (early vs late) is a postoperative, time-dependent outcome, not a baseline prognostic factor. Because it occurs after resection and after the initiation of the risk period for DFS and OS, incorporating it into survival analyses without a dedicated landmark or time-dependent design would introduce time-dependent bias and violate the assumptions of standard Cox proportional hazards modeling. Such outcome-derived variables are not appropriate predictors in a preoperative or postoperative baseline prognostic model. Second, the observation that patients with early recurrence have poorer survival is well-established in the iCCA and broader HPB literature, reflecting aggressive tumor biology rather than providing new prognostic insight. A simple dichotomization using an arbitrary cutoff (such as 365 days) does not add meaningful clinical value and may create redundancy rather than enhance the interpretability of the study. For these statistical and clinical reasons, the early/late recurrence analysis was removed from the Results. We acknowledge that a sentence referring to early recurrence remained in the Abstract and created inconsistency. This statement has now been removed, and the Abstract has been revised to focus on pathological morphology, which is a recognized postoperative classifier and consistent with prior publications. We thank the reviewer for helping us clarify and improve the coherence of the manuscript
- The conclusion implies that the systemic inflammatory response is related to nutrition but the abstract does not provide data to support this interpretation.
Author Respond Q5: We thank the reviewer for this comment. We have revised the conclusion to align more closely with the data presented in the abstract and to avoid overinterpretation regarding the relationship between systemic inflammation and nutritional status.
“Conclusion: Preoperative NLR was associated with poorer DFS and OS in iCCA patients undergoing curative-intent resection. This association was consistently observed in subgroups with R0 resection and in those who developed recurrence. Meanwhile, the CONUT score showed limited independent significance only among patients with R0 resection who experienced recurrence.”
Introduction
Absence of key issues;
- The novelty of this study. What exactly has not previously been analysed?
Author Response Q1: Thank you for this important comment regarding the novelty of the study. We have revised the Introduction to clearly articulate the specific gaps in the existing literature and to clarify the novel aspects of our work. In particular, we now emphasize that prior studies have largely evaluated preoperative inflammatory or nutritional indices individually and in overall cohorts only, with limited attention to clinically relevant subgroups. We further clarify that, to date, no study has systematically compared multiple preoperative inflammatory and immuno-nutritional indices within the same iCCA population or assessed whether their prognostic utility persists across different disease-course contexts, including after recurrence. The study objectives have been revised accordingly to align with these clarified novel aspects.
Introduction at line 85-106
“Despite the growing interest in preoperative inflammatory and immuno-nutritional biomarkers, robust comparative evidence remains limited regarding their relative prognostic performance for DFS and OS, particularly within clinically relevant subgroups. These include patients achieving margin-negative (R0) resection, those who subsequently develop recurrence, and patients with R0 resection who later experience recurrence. To date, no study has simultaneously compared multiple preoperative inflammatory and nutritional indices within the same population or systematically evaluated whether their prognostic utility persists across different disease-course contexts, including the post-recurrence setting.
Therefore, the objectives of this study were first to compare the prognostic associations of established preoperative inflammatory and immuno-nutritional indices, including NLR, LMR, PNI, and CONUT, with disease-free survival and overall survival in the overall cohort of patients with iCCA undergoing curative-intent hepatic resection. We subsequently evaluated the prognostic performance of these indices in clinically relevant subgroups, including patients achieving margin-negative (R0) resection, those who developed recurrence, and patients with R0 resection who subsequently experienced recurrence, in order to assess the consistency and context-specific utility of these commonly used preoperative scores across different surgical and disease-course settings. Given that these biomarkers are inexpensive, non-invasive, and routinely available in clinical practice, this approach aims to inform preoperative risk stratification and prognostic assessment in iCCA, while emphasizing that such indices should not be used as determinants of surgical eligibility.”
- Can prehabilitation reverse the immune and nutritional profiles and improve outcome.?
Author Response Q2: We thank the reviewer for this important and clinically relevant question. In the present retrospective study, data on prehabilitation were not systematically collected and therefore could not be formally analyzed. However, this does not imply that preoperative optimization was not considered in routine clinical practice. During the study period, nutritional support and physical conditioning were addressed on an individualized basis at the discretion of the treating physicians, according to each patient’s clinical condition.
As these interventions were not implemented as part of a standardized prehabilitation program and were not consistently documented with respect to indication, duration, or intensity, their potential effects on immune or nutritional profiles and postoperative outcomes could not be quantitatively assessed.
We agree that this represents an important and clinically relevant aspect, and we are happy to explicitly acknowledge this point as a limitation of our study and as an area for future improvement and investigation.
Add to discussion in line 618-620:
“In addition, standardized data on preoperative prehabilitation were not available, preventing assessment of its potential effects on immune-nutritional status and clinical outcomes.”
- Have any thresholds been identified to suggest levels of biomarkers that should / would contraindicate resection of iCCA or other cancers?
Author Response Q3: Thank you for this important question. To our knowledge, no validated thresholds of preoperative inflammatory or immuno-nutritional biomarkers (such as NLR, LMR, PNI, or CONUT) have been established to contraindicate curative-intent resection in iCCA or other solid tumors. In routine clinical practice, the decision to proceed with hepatic resection relies on a comprehensive multidisciplinary assessment integrating imaging-based resectability, tumor extent, functional hepatic reserve, overall laboratory parameters, performance status, comorbidities, and physiological fitness for major resection. Within this framework, immune–inflammatory and nutritional scores function only as adjunctive risk-stratification tools and should not be used as stand-alone criteria to deny or permit resection. We have clarified this point in the revised manuscript and emphasize that our findings support the use of these indices for prognostic stratification, not as thresholds to contraindicate resection.
Introduction in line 72-84:
“In cancer research, particularly in iCCA, no universally accepted or clinically validated cut-off values exist for these inflammatory or immuno-nutritional indices. Reported thresholds vary substantially across studies, with NLR values in the approximate range of 1.93 to 5 frequently associated with adverse survival outcomes and an increased risk of recurrence, depending on the study population and analytical approach [14,15,18,19]. Likewise, although the CONUT score provides categorized ranges that may facilitate nutritional assessment, these scores have not been validated to contraindicate curative-intent resection in routine clinical practice. Accordingly, NLR, LMR, PNI, and CONUT should be regarded as adjunctive tools for perioperative risk stratification and postoperative management rather than determinants of surgical eligibility. Moreover, these indices may help identify patients who could benefit from prehabilitation or nutritional optimization before resection, potentially contributing to improved DFS and OS.”
Discussion in 567-577:
“It is important to emphasize that the decision to perform curative-intent hepatic resection for iCCA is fundamentally based on multidimensional clinical evaluation rather than any single biomarker. Operability depends on imaging-based resectability including vascular involvement, biliary extension, and the feasibility of achieving an adequate future liver remnant along with hepatic functional reserve, physiological fitness, performance status, and comorbidities. Consequently, inflammatory and immuno-nutritional biomarkers such as NLR, and CONUT should be regarded as adjunctive prognostic tools rather than determinants of surgical eligibility. To date, no validated biomarker thresholds exist that would contraindicate curative-intent resection in iCCA or other solid tumors. These indices therefore support perioperative risk stratification but should not be used as stand-alone criteria to deny or permit resection.”
Methods
- If the study is specific to iCCA why was the wider cohort with pCCA and dCCA mentioned. They should be excluded or the reason for inclusion identified.
Author Response Q1: We thank the reviewer for this important comment. The present study was specifically designed to include only patients with intrahepatic cholangiocarcinoma (iCCA). In the original version of the manuscript, the broader cholangiocarcinoma (CCA) cohort was mentioned to describe the source population of patients undergoing curative-intent resection during the study period; however, patients with perihilar (pCCA) and distal cholangiocarcinoma (dCCA) were excluded from the analysis according to the predefined eligibility criteria.
To address the reviewer’s concern and improve clarity, we have revised Section 2.1 (Patients) to explicitly state that only patients with pathologically confirmed iCCA were included in the present analysis. In addition, Figure 1 (study flowchart) has been updated to clearly depict the exclusion of pCCA and dCCA cases and to ensure consistency between the Methods section and the final study cohort.
Method section at line 108-124:
“ 2.1. Patients
This retrospective study included patients who underwent curative-intent resection for intrahepatic cholangiocarcinoma (iCCA) at Srinagarind Hospital, Khon Kaen University, Thailand, between January 2015 and December 2021. During the study period, a total of 220 patients with pathologically confirmed iCCA underwent curative-intent hepatic resection. Patients were selected for the present analysis according to predefined inclusion and exclusion criteria. Inclusion criteria were as follows: (i) curative-intent hepatic resection for iCCA, (ii) histopathological confirmation of iCCA, (iii) had an ECOG performance status of 0–1, (iv) did not receive neoadjuvant chemotherapy, and (v) had complete preoperative laboratory, clinicopathological, and follow-up data. Preoperative tumor biomarkers, including carcinoembryonic antigen (CEA) and carbohydrate antigen 19-9 (CA19-9), were collected when available and used for subsequent analyses. Exclusion criteria were as follows: (i) those with other biliary tract diseases prior to resection, (ii) patients who underwent repeat hepatectomy for recurrent disease, (iii) patients lost to follow-up after resection, and (iv) those with incomplete clinical or laboratory records. After applying these criteria, a total of 213 patients were eligible and included in the final analysis (Figure 1).”
- Preoperative nutrition was given. The duration and reason should be identified and how the calories given correlated with nutritional status.
Author Response Q2: We thank the reviewer for this important comment. This study was retrospective in nature, and detailed information regarding preoperative nutritional support, including indications, duration, and caloric intake, was not systematically recorded and therefore not available for review.
During the study period, preoperative nutritional care was individualized and provided according to routine clinical practice and the treating physicians’ judgment based on each patient’s clinical condition, rather than following a standardized research-specific preoperative nutritional or prehabilitation protocol. As a result, information on the rationale for nutritional intervention, duration of support, and caloric intake could not be reliably retrieved or quantitatively analyzed, and correlations with nutritional indices such as PNI or CONUT could not be assessed.
We acknowledge this as a limitation of the present retrospective analysis. Future prospective studies incorporating standardized, protocol-driven preoperative nutritional and prehabilitation strategies with systematic documentation of caloric intake may allow a more detailed evaluation of the impact of preoperative nutrition on immune-nutritional status and clinical outcomes.
Discussion in limitation part at line 616-618:
“In addition, standardized data on preoperative prehabilitation were not available, preventing assessment of its potential effects on immune-nutritional status and clinical outcomes.”
- Pre-op modulation? eg biliary drainage, residual liver measurement, PVE/HVE utilized.
Author Response Q3: We appreciate the reviewer’s comment regarding preoperative modulation strategies. The present study focused exclusively on patients with intrahepatic cholangiocarcinoma (iCCA). In routine clinical practice, iCCA is less frequently associated with obstructive jaundice compared with perihilar cholangiocarcinoma; therefore, preoperative biliary drainage is not routinely indicated or performed in the majority of patients with iCCA in our cohort.
Similarly, preoperative portal vein embolization (PVE) or hepatic vein embolization (HVE) was not routinely utilized in this iCCA cohort during the study period and was more commonly applied in patients with perihilar cholangiocarcinoma. Assessment of the future liver remnant was performed selectively as part of routine surgical planning based on anatomical considerations and liver function. However, due to the retrospective nature of the study and the earlier treatment period, detailed quantitative data on residual liver volume were not consistently available for inclusion in the present analysis.
- Was there a specialist MDT.? What was the pre-op radiology TNM stage for the cohort.
Author Response Q4: We thank the reviewer for this important comment. During the study period, patients were routinely discussed in a multidisciplinary team (MDT) setting involving hepatobiliary surgeons, radiologists, and medical oncologists to guide treatment decisions on a case-by-case basis. Preoperative evaluation included cross-sectional imaging (CT and/or MRI) together with relevant clinical and laboratory parameters to assess resectability and operative risk prior to curative-intent resection.
However, although preoperative imaging was systematically used for clinical decision-making, formal documentation of preoperative radiologic TNM staging was not consistently recorded in this retrospective dataset and therefore could not be reliably extracted or included in the present analyses. As a result, preoperative TNM stage was not incorporated into the statistical models.
We acknowledge that incorporation of standardized preoperative radiologic staging represents an important concept and may further improve preoperative risk stratification and prognostic prediction. This will be an important consideration for future prospective studies with more comprehensive and standardized data collection.
- What was the purpose of the growth pattern evaluation by a non pathologist?
Author Response Q5: We thank the reviewer for this important comment and appreciate the opportunity to clarify this point. In routine clinical practice, macroscopic growth patterns of iCCA (mass-forming, periductal-infiltrating, and intraductal types) are assessed at different stages of patient management and for different purposes.
In the preoperative setting, growth pattern assessment based on imaging findings is typically performed by radiologists in conjunction with hepatobiliary surgeons. This evaluation reflects the distinct natural history and growth behavior of each subtype and assists in surgical planning. For example, intraductal and periductal-infiltrating growth patterns often necessitate particular attention to bile duct margins, whereas mass-forming tumors predominantly involve the hepatic parenchyma and require careful assessment of parenchymal resection margins. These imaging-based assessments are therefore used to evaluate disease extent, biological aggressiveness, and surgical resectability, thereby informing individualized operative strategies prior to resection.
In contrast, postoperative macroscopic growth pattern evaluation, which serves as a prognostic indicator, is based on histopathological examination and is assessed by pathologists. Importantly, in the present study, macroscopic growth patterns were derived exclusively from postoperative pathological reports and were evaluated by pathologists. These data were used to characterize tumor biology and prognosis and were not based on preoperative imaging assessments by non-pathologists.
We have clarified this distinction in the revised manuscript to avoid potential misunderstanding regarding the source and purpose of growth pattern evaluation in this study.
- Unusual to label all deaths to be due to cancer recurrence, even when no radiological evidence of recurrence. Defining the nature of the terminal event could lead to a more accurate assessment of recurrence.
Author Response Q6: We thank the reviewer for highlighting this important methodological consideration. We agree that clear and precise definition of survival endpoints is essential for accurate outcome assessment in retrospective studies. In response to this comment, we have revised the definition of disease-free survival (DFS) in the Methods section to improve methodological clarity and consistency with commonly accepted practices in survival analysis. Specifically, DFS is now defined as the time from liver resection to the first documented recurrence of iCCA or death from any cause, whichever occurred first, or last follow-up. Accordingly, the previous statement indicating that deaths without documented recurrence were assumed to be cancer-related has been removed from the manuscript. This revision minimizes the potential for misclassification and provides a more robust and transparent assessment of patient outcomes.
Method at line 192-195:
“2.5. Postoperative Follow-up and Outcome measurements
DFS was defined as the time from liver resection to the first documented recurrence of iCCA or death, whichever occurred first, or last follow-up, consistent with previous surgical oncology studies [25,26]. Recurrence was determined based on histopathological confirmation, radiologic evidence from ultrasound, CT, or MRI, or clinical documentation in the medical records.”
Results
- Should be labelled as potentially curative resection as the majority recur.
Author Response Q1: We thank the reviewer for this important comment. We respectfully prefer to retain the term “curative-intent resection”, as this terminology is widely used and well established in surgical oncology to describe resections performed with the intention of complete tumor removal (R0 resection) at the time of resection. The term refers to the therapeutic intent at the time of treatment, rather than the eventual oncologic outcome.
- Median age of cohort confusing as appears to be applied to a sub group.
Author Response Q2: We thank the reviewer for this helpful comment. We acknowledge that the median age was previously described in a way that could be misinterpreted as applying to a subgroup. We have revised the manuscript to clearly state that the median age refers to the entire study cohort, reported as a continuous variable. This information has been clarified consistently in both the text and Table 1, and age stratification is now described separately to avoid ambiguity.
- BMI should be given as median and range
Author Response Q3: We thank the reviewer for this comment. BMI has been revised to be reported as a continuous variable using the median and range in both the text and Table 1. Where applicable, BMI stratification is now presented separately to avoid ambiguity.
- NLR, LMR, PNI and CONUT have been grouped as low or high based on median values. This produces only two cohorts and does not allow useful prognostic information for individual patients which would need continuous variables to be correlated with outcome.
Author Response Q4: We thank the reviewer for this helpful comment. We agree that dividing continuous markers into high and low groups makes interpretation easier but may limit their use for individualized prognosis. The main aim of this study was to compare the prognostic value of established preoperative inflammatory and nutritional scores at the group level, rather than to build an individualized prediction model.
All scores were therefore also examined as continuous variables in Cox regression analyses. This limitation has now been clearly acknowledged in the Discussion, and we have indicated in the Future Directions section that future prospective studies using continuous biomarker analysis will be needed to improve individualized risk assessment.
- No previously validated cut off values for theprognostic markers have been applied.
Author Response Q5: We thank the reviewer for this important comment. We acknowledge that there are currently no universally accepted or consistently validated cut-off values for preoperative inflammatory and immunonutritional markers such as NLR, LMR, and PNI in intrahepatic cholangiocarcinoma. Previous studies have applied heterogeneous approaches to define cut-off values, including ROC-based methods (Youden index), median-based stratification, and other data-driven thresholds.
In the present study, cohort-specific median values were therefore applied for NLR, LMR, and PNI to allow internal risk stratification while minimizing overfitting in this retrospective cohort. Importantly, the resulting cut-off values were comparable to those reported in prior studies.
In contrast, the CONUT score is a clinically established nutritional assessment tool with predefined severity categories. Accordingly, patients were classified based on standard CONUT grading, with moderate to severe malnutrition defined as the high-risk group, consistent with previous surgical oncology studies. We have clarified this rationale in the
Methods section in line 181-184.
“2.4. Preoperative score assessment
All scores were analyzed both as continuous variables and as categorical variables, dichotomized into high and low groups based on previously validated cutoff values or the median values, as appropriate, whereas CONUT was classified based on established severity categories as described in our previous study [17].”
- The NLR high vs low values show a striking association with survival and the data excluding confounders such as involved margins and outcome of those with recurrence of interest.
Author Response Q6: We thank the reviewer for this positive and insightful comment. We agree that the consistent association between preoperative NLR and survival, even after accounting for important clinical confounders such as surgical margin status and recurrence, supports the robustness of NLR as an independent host-related prognostic marker in patients with iCCA.
- Would be useful to analyse those with recurrence both from baseline and from diagnosis of recurrence.
Author Response Q7: We thank the reviewer for this thoughtful and clinically relevant suggestion. We agree that analyzing outcomes in patients with recurrence both from the time of resection and from the time of recurrence diagnosis could provide additional insights into disease course and prognosis. However, in the present retrospective cohort, detailed and standardized information regarding post-recurrence management, including systemic therapy, locoregional treatment, repeat resection, or supportive care, was not consistently available. As post-recurrence treatment is a major determinant of survival after recurrence, the lack of these data may introduce unmeasured confounding and limit the reliability of post-recurrence survival analyses. For this reason, we did not perform survival analyses from the time of recurrence in the current study. We acknowledge this as an important limitation and believe that this represents a valuable direction for future prospective studies with systematic collection of post-recurrence treatment data.
Discussion in limitation
“Finally, post-recurrence treatments, including chemotherapy, radiotherapy, and redo resection, were incompletely documented and could not be reliably categorized or analyzed, and thus were not included in the present study.”
- Conclusion to results should be that a correlationhas been shown with outcome in resected iCCA but there is no evidence that it determined outcome as suggested by authors. This would require a mechanistic rather than observational study.
Author Response Q8: We thank the reviewer for this important and constructive comment. We agree that the present study demonstrates an association between preoperative NLR and survival outcomes in patients with resected intrahepatic cholangiocarcinoma, rather than a causal or mechanistic relationship. Accordingly, we have revised the Conclusions and relevant sections of the manuscript to avoid causal language and to clearly reflect the observational nature of the study.
We also acknowledge that establishing whether NLR directly determines clinical outcomes would require mechanistic investigations beyond the scope of the current observational analysis. We have clarified this point in the revised manuscript and highlighted the need for future mechanistic studies to further elucidate the biological pathways underlying these associations.
Result of section 3.5 at line 432-435:
“These findings demonstrate that both CONUT score and tumor morphology are independently associated with post-recurrence outcomes, with impaired nutritional condition (as reflected by moderate–severe CONUT) and non-ID tumor types being significant prognostic factors for poor overall survival in margin-free resections with recurrence.”
Conclusion section at line 645-661:
“This study demonstrates that elevated preoperative NLR is an independent and consistent factor associated with poorer DFS and OS in patients with iCCA undergoing curative-intent resection. The prognostic association of NLR persisted across clinically relevant subgroups, including patients who achieved margin-negative resection and those who subsequently developed recurrence, supporting its value as a broadly applicable prognostic marker reflecting adverse tumor biology and systemic inflammatory status. In contrast, the association of the CONUT score with outcome was limited and context dependent, with independent significance observed only in patients with margin-negative resection who later experienced recurrence, suggesting a selective prognostic association of nutritional status in specific clinical settings. Taken together, these findings indicate that preoperative systemic inflammation is strongly associated with long-term prognosis, whereas nutritional impairment may exert additional prognostic influence only in selected patient subsets. Accordingly, NLR may be useful for general preoperative risk stratification, while the CONUT score may provide supplementary prognostic information in specific clinical contexts. However, as an observational study, these findings demonstrate correlation, and further mechanistic studies are warranted to explore whether targeted interventions can influence these outcomes.”
- Notable absences from data analysis
Author Response Q9: We acknowledge the reviewer’s point regarding notable absences from the data analysis. As described in the Limitations section, several clinically relevant preoperative variables, including imaging-based tumor characteristics and serum tumor biomarkers, were not consistently available and therefore could not be included in the multivariable analyses. We agree that the absence of these data limits mechanistic interpretation of the observed associations. This limitation has been clearly highlighted in the revised manuscript and supports a cautious interpretation of the findings, focusing on correlation rather than causation, as reflected in the revised Conclusion.
Discussion in limitation at line 611-619:
“Third, several clinically relevant preoperative variables, including imaging-based tumor characteristics (such as tumor size and macroscopic morphology) and serum tumor biomarkers, were incompletely recorded, precluding their inclusion in the multivariable Cox regression models. This limitation may have restricted evaluation of the independent prognostic contribution of tumor-related factors and may partly explain the predominant significance observed for inflammatory and nutritional indices. In addition, standardized data on preoperative prehabilitation were not available, preventing assessment of its potential effects on immune-nutritional status and clinical outcomes.”
- Effect of adjuvant chemotherapy and /or immunotherapy on cohort survival.?
Author Response Q10: We thank the reviewer for this important and clinically relevant question regarding the effect of adjuvant therapy on survival outcomes.
- With respect to immunotherapy, none of the patients in our cohort received adjuvant immunotherapy, as this modality was not part of the standard treatment protocol for intrahepatic cholangiocarcinoma during the study period.
- Regarding adjuvant chemotherapy, data were collected; however, owing to the retrospective nature of the study and changes in medical record systems over the extended study period, documentation of adjuvant chemotherapy—particularly among patients treated in earlier years—was incomplete, with a substantial proportion of missing data. Given these limitations, adjuvant chemotherapy could not be reliably categorized or incorporated into the multivariable survival analyses without compromising the robustness of the models.
We acknowledge that the absence of this information limits our ability to evaluate the impact of adjuvant chemotherapy on survival outcomes. This limitation has been explicitly addressed in the revised manuscript to ensure appropriate interpretation of the prognostic findings.
Discussion in limitation at line 619-624:
“Fourth, information on perioperative and adjuvant treatments, including adjuvant chemotherapy, was not systematically documented during the earlier study period, limiting the ability to account for their potential impact on survival outcomes. Similarly, owing to the retrospective design and interval-based routine surveillance, some imprecision in determining the exact timing of recurrence was unavoidable, particularly during long-term follow-up.”
- Did prognostic scores normalize following recovery from the iCCA resection ie were the parameters NLR, LMR, PNI, CONUT secondary to the iCCA per se.
Author Response Q11: We thank the reviewer for this thoughtful and clinically insightful comment. Longitudinal assessment of inflammatory and nutritional indices following iCCA resection may indeed provide important insights into whether preoperative abnormalities in NLR, LMR, PNI, and CONUT are primarily tumor-driven or reflect underlying host-related factors.
However, postoperative measurements of these indices were not systematically collected in the present study, and longitudinal normalization patterns could therefore not be evaluated. We agree that investigating whether elevated preoperative scores normalize after tumor removal—or persist despite resection—represents a clinically relevant and biologically meaningful question. Such analyses may help distinguish reversible tumor-related inflammation and malnutrition from more persistent host-related conditions and may further refine postoperative risk stratification.
This concept represents an important direction for future research and will be explored in prospective studies with standardized postoperative laboratory assessments. We have acknowledged the absence of longitudinal postoperative data as a limitation of the present study.
Discussion in limitation at line 627-631
“Sixth, longitudinal postoperative measurements of inflammatory and nutritional indices were not available; therefore, it was not possible to determine whether preoperative abnormalities in NLR, LMR, PNI, and CONUT normalized after tumor resection or persisted, limiting interpretation regarding tumor-driven versus host-related effects.”
Discussion
Para 1: Authors should identify the novel findings of their study in comparison to prior reports.
Author Response Q1: We thank the reviewer for this constructive comment. We agree that the novel contributions of the present study should be clearly articulated in comparison with prior reports. Accordingly, we have revised the first paragraph of the Discussion to explicitly highlight the unique aspects of our work.
Specifically, we now emphasize that, in contrast to previous studies that primarily evaluated individual inflammatory or nutritional indices in unselected cohorts, the present study provides a direct comparative assessment of multiple preoperative immune-inflammatory and immuno-nutritional indices within the same iCCA population. We further clarify the novelty of evaluating these indices across clinically meaningful post-surgical subgroups, including patients achieving margin-negative (R0) resection and those who subsequently developed recurrence.
These revisions aim to more clearly delineate the added value of our findings relative to prior literature and to strengthen the contextual interpretation of our results.
Discussion at line 451-466:
“This study represents one of the largest single-center analyses to evaluate the prognostic relevance of both a systemic inflammatory index (NLR) and a nutritional index (CONUT score) in patients undergoing curative-intent resection for intrahepatic cholangiocarcinoma (iCCA). Although associations between preoperative inflammatory indices and adverse prognosis in iCCA have been reported previously, the key novel contribution of the present study lies in the comparative assessment of these two distinct host-related factors across clinically meaningful post- surgical subgroups. We observed that elevated preoperative NLR was consistently associated with poorer disease-free survival (DFS) and overall survival (OS), and that this association persisted across important clinical subsets, including patients who achieved margin-negative (R0) resection and those who subsequently developed recurrence. In contrast, the prognostic association of the CONUT score appeared more selective and context dependent, with independent significance observed only in the subgroup of patients with R0 resection who experienced recurrence. Collectively, these findings indicate that preoperative inflammatory and nutritional indices are associated with long-term outcomes in resected iCCA, with NLR demonstrating more consistent prognostic relevance across patient subgroups.”
Para 2: This is background and not findings from this study, Should be moved to introduction or removed.
Author Response Q2: We appreciate the reviewer’s comment. We have revised the paragraph to more explicitly link the background mechanisms to the findings observed in our cohort, and have streamlined the text to avoid general background description in the Discussion.
Para 3: Authors suggest immune/nutritional parameters used in protocol for risk stratification and pre-op planning. Need to justify the use of median values to form two groups for comparison and to discuss the major limitations of this approach for prognostic assessment of an individual. How the grouped data would be applied is unclear.
Author Response Q3: We thank the reviewer for this insightful methodological comment. We agree that dichotomizing continuous immune-inflammatory and immuno-nutritional parameters using median values has important limitations, particularly with respect to individualized prognostic assessment.
In the present retrospective cohort, median-based cut-off values were applied as a pragmatic and commonly used approach to facilitate balanced group-level comparisons in the absence of universally accepted or clinically validated thresholds for NLR, LMR, and PNI in intrahepatic cholangiocarcinoma. This approach was intended for cohort-level risk stratification rather than individualized clinical prediction.
We have revised the manuscript to explicitly justify the use of median cut-offs, to clarify the limitations of this dichotomization for individual patient prognostication, and to emphasize that the clinical utility of these grouped parameters lies in identifying relative risk categories within a surgical population. Accordingly, these indices are best applied as adjunctive tools to inform perioperative risk stratification and planning, in conjunction with established clinicopathological and imaging-based factors, rather than as determinants of individualized clinical decision-making.
Discussion at line 467-483:
“In iCCA, preoperative scores reflecting inflammation and nutritional status provide practical tools for risk stratification and perioperative planning at the population level [14-17]. In this study, we evaluated NLR, LMR, PNI, and CONUT for their prognostic associations with disease-free survival (DFS) and overall survival (OS). In the absence of universally accepted clinical cut-off values for NLR, LMR, and PNI in iCCA, cohort-specific median values (2.4, 3.6, and 52, respectively) were applied to facilitate balanced group-level comparisons. In contrast, CONUT was categorized using standardized thresholds based on serum albumin, total lymphocyte count, and total cholesterol, classifying patients as normal (0–1), mild (2–4), or moderate/severe (≥5) [26]. These indices capture complementary aspects of host status: NLR and LMR reflect systemic inflammation and immune suppression [23,24]; PNI represents combined nutritional and immune status [25]; and CONUT provides an integrated assessment of overall nutritional condition [26]. As non-invasive and cost-effective measures, these indices support cohort-level preoperative risk stratification and inform perioperative planning, rather than individualized clinical decision-making. Subgroup analyses in patients achieving R0 resection and those who developed recurrence further clarified the context-specific prognostic relevance of these commonly used scores.”
Limitation at line 624-627:
“Fifth, although inflammatory and immunonutritional indices were dichotomized using cohort-specific cut-off values to facilitate clinical interpretability and group-level risk stratification, this approach may have reduced granularity and limited individualized prognostic assessment.”
Para 4: Discussion should focus on how the indices could be applied to iCCA rather than discussion of other disease groups where the prognostic indices correlate with survival.
Author Response Q4: We thank the reviewer for this comment. We have revised the Discussion to focus specifically on the application of inflammatory and nutritional indices in intrahepatic cholangiocarcinoma (iCCA). The revised text emphasizes the prognostic relevance of preoperative NLR for cohort-level risk stratification in iCCA patients undergoing curative-intent resection, including consistency across clinically relevant subgroups (R0 resection and recurrence). Comparisons with other disease entities have been minimized, and external literature is now cited primarily to contextualize NLR cut-off values within previously reported iCCA-specific studies.
Discussion at line 484-515:
“Within this framework of cohort-level risk stratification, our study demonstrates that an elevated preoperative neutrophil-to-lymphocyte ratio (NLR ≥ 2.4) is independently associated with poorer disease-free survival (DFS) and overall survival (OS) in patients undergoing curative-intent resection for iCCA. This association remained consistent in the overall cohort (DFS: HR = 1.66; OS: HR = 1.94) and persisted in clinically relevant subgroups, including patients achieving margin-negative (R0) resection (DFS: HR = 1.66; OS: HR = 2.11) and those who subsequently developed recurrence (OS: HR = 1.83). These findings highlight the prognostic relevance of NLR across the disease course of iCCA, even among patients with apparently favorable surgical outcomes. Compared with other inflammation- and nutrition-based indices evaluated in this study, including LMR, PNI, and CONUT, NLR demonstrated the most consistent association with survival outcomes. With regard to prognostic thresholds, the NLR cut-off value of 2.4 observed in this cohort lies within the range commonly reported in prior iCCA studies, where values between approximately 2–4 have been associated with adverse survival outcomes and increased recurrence risk [14,15,19]. These observations are further supported by a systematic review and meta-analysis including 15 studies (18 cohorts) with a total of 4,123 cases with iCCA, which showed that elevated preoperative NLR using cut-off values ranging from 1.93 to 5 was significantly associated with shorter OS and RFS [18].”
Para 5. Optimal scores have not been determined as suggested, only cancer risks associated with low or high scores. How would cohort data be translated into some clinical value for individual patient management.
Author Response Q5: We thank the reviewer for this important comment. We agree that optimal prognostic thresholds for individual patient decision-making cannot be determined from the present observational cohort. Accordingly, we have revised the Discussion to clarify that the identified NLR threshold represents a cohort-level risk stratification point, rather than an optimal or definitive clinical cut-off. We now emphasize that while NLR may help identify patients at relatively higher risk who could benefit from closer postoperative surveillance or consideration of adjuvant strategies, translation of these cohort-based associations into individualized patient management requires prospective validation, standardized thresholds, and integration with established clinicopathological and imaging-based factors.
Discussion at line 507-515:
"Importantly, the identified NLR threshold should not be interpreted as an optimal or definitive cut-off for individual patient decision-making. Rather, it represents a cohort-level risk stratification point associated with differential survival outcomes. Clinically, NLR may help identify patients at relatively higher risk who could benefit from closer postoperative surveillance or consideration of adjuvant strategies when clinically appropriate. However, translation of these cohort-based associations into individualized patient management requires prospective validation, standardized thresholds, and integration with established clinicopathological and imaging-based factors."
Para 6: Mechanistic discussion important but current study has not contributed to our understanding of mechanism. CONUT discussion important but authors should clarify the numbers in the moderate/ severe group in whom survival is reduced. Again no thresholds identified which could be used in clinical practise.
Author Response Q6: We thank the reviewer for this insightful comment regarding the mechanistic discussion. We acknowledge that the present study was observational in nature and did not directly investigate biological mechanisms. Accordingly, we have revised this section to soften the interpretive tone, reduce speculative statements, and clearly indicate that the proposed mechanisms are derived from previously published literature and are intended to provide biological context rather than mechanistic evidence from this cohort.
Discussion at line 516-540:
“These findings support the potential role of NLR as a prognostic biomarker reflecting aggressive tumor behavior and an unfavorable disease course in iCCA. NLR is widely regarded as an indicator of the balance between systemic inflammation and host immune response. An elevated NLR reflects a relative increase in circulating neutrophils and/or a reduction in lymphocytes, a profile that has been associated with tumor-promoting conditions and adverse clinical outcomes. Prior studies have shown that neutrophils can facilitate tumor progression through the secretion of pro-inflammatory cytokines, growth factors, and matrix-degrading enzymes, thereby promoting angiogenesis, invasion, and metastasis [38,39]. In addition, neutrophil-derived reactive oxygen species and arginase have been reported to suppress lymphocyte proliferation and cytotoxic activity, contributing to impaired anti-tumor immunity [40,41]. In contrast, lymphocytes, particularly cytotoxic T cells and natural killer cells, play a central role in tumor surveillance and immune-mediated tumor control. A reduced lymphocyte count, as reflected by a high NLR, has therefore been associated with compromised adaptive immunity and diminished tumor control [42]. Taken together, these biologically plausible mechanisms described in prior literature may help explain why elevated preoperative NLR has been consistently associated with poorer DFS and OS across multiple cancer types, including iCCA [28,30,36,43]. Notably, although patients achieving margin-negative (R0) resection are generally expected to have favorable outcomes, the occurrence of recurrence in a subset of these patients may reflect underlying aggressive tumor biology that is not fully captured by conventional pathological assessment [44]. It should be emphasized that the present study was observational in nature and did not directly investigate these biological mechanisms; therefore, the mechanistic interpretations discussed here are based on established evidence from prior studies rather than direct experimental validation within this cohort.”
Para 7: Some insight into limitation of study but no clarity on how these findings will be applied to patients selection for resection, prehabilitation, nutrition supplementation or post op management.
Author Response Q7: We thank the reviewer for this important and constructive comment regarding the clinical applicability of our findings. In response, we have revised the Discussion to more clearly clarify how these biomarkers may be applied in clinical practice. Specifically, we now emphasize that decisions regarding curative-intent hepatic resection for iCCA are based on comprehensive clinical assessment, including imaging-based resectability, hepatic functional reserve, and patient fitness, rather than any single biomarker. Accordingly, inflammatory and immuno-nutritional indices such as NLR and CONUT are described as adjunctive prognostic tools rather than determinants of surgical eligibility, and we explicitly state that no validated biomarker thresholds currently exist to contraindicate curative-intent resection.
We further clarify that, although these biomarkers should not be used to select or exclude patients from resection, their prognostic information may have practical implications for perioperative management, including postoperative risk stratification, intensity of surveillance, consideration of adjuvant therapy, and nutritional optimization. We also discuss the potential role of prehabilitation and nutritional support as future applications, while acknowledging that such interventions were not systematically assessed in the present retrospective study and warrant evaluation in prospective investigations. These revisions have been incorporated to better delineate the clinical relevance and appropriate limitations of our findings.
Discussion at line 566-591:
“It is important to emphasize that the decision to perform curative-intent hepatic resection for iCCA is fundamentally based on multidimensional clinical evaluation rather than any single biomarker. Operability depends on imaging-based resectability including vascular involvement, biliary extension, and the feasibility of achieving an adequate future liver remnant along with hepatic functional reserve, physiological fitness, performance status, and comorbidities. Consequently, inflammatory and immuno-nutritional biomarkers such as NLR, and CONUT should be regarded as adjunctive prognostic tools rather than determinants of surgical eligibility. To date, no validated biomarker thresholds exist that would contraindicate curative-intent resection in iCCA or other solid tumors. These indices therefore support perioperative risk stratification but should not be used as stand-alone criteria to deny or permit resection.
Although these biomarkers should not determine surgical eligibility, their prognostic information has practical implications for perioperative management. Patients presenting with markedly elevated NLR or impaired nutritional indices may be considered a higher-risk subgroup who could benefit from closer postoperative surveillance, more intensive follow-up imaging, or prioritization for adjuvant systemic therapy when appropriate. Similarly, identifying poor nutritional reserve through CONUT may help guide targeted nutritional counseling or optimization before major hepatectomy, even though such measures are unlikely to reverse tumor-driven inflammation. In this context, structured prehabilitation strategies incorporating nutritional optimization and physical conditioning may represent a potentially beneficial future application to enhance perioperative resilience and postoperative recovery. However, such interventions were not systematically assessed in the present retrospective cohort and should be evaluated in future prospective studies. Thus, the clinical value of these indices lies in risk stratification and individualized perioperative planning rather than modifying or delaying resection based on biomarker normalization.”
Figures: Helpful and clear
Author Response: We thank the reviewer for this positive comment. We are pleased that the figures were considered helpful and clearly presented.
Tables: Require explanation of abbreviations in foot notes.
Author Response: We thank the reviewer for this helpful comment. All abbreviations used in Tables 1–3 have now been fully defined in the corresponding table footnotes, and the formatting has been standardized throughout the manuscript.
References: Relevant and up to date
Author Response: We appreciate the reviewer’s positive assessment of the references.

Reviewer 2 Report
Comments and Suggestions for Authors
Summary:
This manuscript investigates the prognostic role of preoperative inflammation and immunonutritional factors, particularly neutrophil-to-lymphocyte ratio (NLR), lymphocyte-to-monocyte ratio (LMR), prognostic nutritional index (PNI), and CONUT score, for patients with intrahepatic cholangiocarcinoma (iCCA) undergoing curative resection. This issue has practical relevance, since the use of preoperative prognosticators is a critical consideration for the treatment of this very aggressive tumor. In fact, the study not only provides subgroup analyses, focusing on cases with margin-negative (R0) resection, but also on patients with recurrence, adding valuable insight into the results. Specifically, it appears that the CONUT score is a prognostic indicator for survival in recurring patients.
However, it is apparent that the document also demonstrates contradictions between the narrative provided and the data reported, particularly with regards to which variables had statistically significant results for the univariate analysis. There are also inconsistencies between the variables that figure within the range of the study parameters, specifically PLR, and the variables actually reported. These need to be addressed for the scientific integrity of the study results.
Specific Comments:
- In the introduction section, the objective stated is to evaluate “NLR, PLR, PNI, and CONUT scores.” However, PLR is completely absent from the Abstract and the Results tables (Tables 1-7). If the analysis of PLR was performed, the data should be included. However, if the data was not included, references to PLR should not appear in the Introduction and Discussion.
- Section 3.1., the text refers to “NLP”. Please clarify if NLP is a typo for another variable and if the value is correct.
- Section 3.2., "Although LMR, PNI, and CONUT showed significance in univariate analysis.", This goes against the date in Tables 2 and 3. Based on the tables you provided, PNI was not significant. Please make the appropriate corrections regarding the data.
- The study uses median values as the cut-off points. While acceptable, the authors should have explained why they used the median score, when ROC curve analysis for the Youden index is usually the preferred method for getting the optimal cut-off points.
- Lines 448-448, “NLR demonstrated superior prognostic performance compared to ……. ,including the PLR." Since the PLR data is not shown in the results, this claim should be revised.
Author Response
Overall response and summary of revisions
We sincerely thank the Editor and all reviewers for their constructive and insightful comments, which improved the clarity, methodological transparency, and clinical interpretation of our manuscript entitled “Preoperative Prognostic Score for Patients with Intrahepatic Cholangiocarcinoma Undergoing Curative-Intent Resection.” We have addressed all points and revised the manuscript accordingly. The major revisions include: (1) clarification of cohort definition and surgical intent, (2) correction of terminology and internal inconsistencies, (3) strengthening of the Methods and endpoint definitions, (4) improved presentation and consistency of Results and tables, and (5) refinement of the Discussion/Conclusion to avoid causal language and emphasize cohort-level risk stratification and appropriate clinical applicability. Detailed point-by-point responses are provided below.
Key revisions across the manuscript
- Terminology and cohort definition: Replaced “curative resection” with “curative-intent resection” throughout to reflect intent at resection (with margin and nodal status confirmed postoperatively). Clarified inclusion of R1 resections and resected nodal metastases when no gross residual disease remained, ensuring consistency across Abstract/Methods/Results.
- Methods and outcomes: Clarified that only iCCA cases were included and updated Figure 1 to show exclusions of pCCA/dCCA. Revised DFS definition to standard practice (time to recurrence or death from any cause) and removed the assumption that deaths without recurrence were cancer-related. Clarified recurrence ascertainment using routine surveillance and acknowledged that recurrence timing reflects first detection.
- Data consistency and corrections: Removed erroneous references to PLR (not analyzed), corrected “NLP” to “NLR,” and revised statements (univariate significance for PNI) to match the tables.
- Results presentation: Clarified reporting of age to avoid subgroup misinterpretation, revised BMI reporting (median and range), and added complete abbreviation definitions in table footnotes.
- Cut-offs and limitations: Justified the use of median-based cut-offs for NLR/LMR/PNI in the absence of validated iCCA thresholds, and explicitly noted that dichotomization supports cohort comparisons but limits individualized prognostic inference (with continuous-variable analyses addressed where applicable).
- Clinical interpretation: Revised Discussion/Conclusion to emphasize association (not causation) and that mechanistic confirmation requires dedicated studies. Clarified that CONUT findings were context-dependent (mainly in the R0 + recurrence subgroup) and that indices should be used as adjunctive tools for cohort-level risk stratification, not determinants of surgical eligibility.
- Limitations and future directions: Acknowledged missing/incomplete data (imaging tumor features, tumor markers, adjuvant/post-recurrence treatment, longitudinal indices) and highlighted the need for prospective studies with standardized data capture and evaluation of prehabilitation/nutrition protocols.
Reviewer 2
Open Review
(x) I would not like to sign my review report
( ) I would like to sign my review report
Quality of English Language
( ) The English could be improved to more clearly express the research.
(x) The English is fine and does not require any improvement.
|
Yes |
Can be improved |
Must be improved |
Not applicable |
|
|
Does the introduction provide sufficient background and include all relevant references? |
( ) |
(x) |
( ) |
( ) |
|
Is the research design appropriate? |
(x) |
( ) |
( ) |
( ) |
|
Are the methods adequately described? |
( ) |
(x) |
( ) |
( ) |
|
Are the results clearly presented? |
( ) |
( ) |
(x) |
( ) |
|
Are the conclusions supported by the results? |
( ) |
(x) |
( ) |
( ) |
|
Are all figures and tables clear and well-presented? |
( ) |
(x) |
( ) |
( ) |
Comments and Suggestions for Authors
Summary:
This manuscript investigates the prognostic role of preoperative inflammation and immunonutritional factors, particularly neutrophil-to-lymphocyte ratio (NLR), lymphocyte-to-monocyte ratio (LMR), prognostic nutritional index (PNI), and CONUT score, for patients with intrahepatic cholangiocarcinoma (iCCA) undergoing curative resection. This issue has practical relevance, since the use of preoperative prognosticators is a critical consideration for the treatment of this very aggressive tumor. In fact, the study not only provides subgroup analyses, focusing on cases with margin-negative (R0) resection, but also on patients with recurrence, adding valuable insight into the results. Specifically, it appears that the CONUT score is a prognostic indicator for survival in recurring patients.
However, it is apparent that the document also demonstrates contradictions between the narrative provided and the data reported, particularly with regards to which variables had statistically significant results for the univariate analysis. There are also inconsistencies between the variables that figure within the range of the study parameters, specifically PLR, and the variables actually reported. These need to be addressed for the scientific integrity of the study results.
Specific Comments:
- In the introduction section, the objective stated is to evaluate “NLR, PLR, PNI, and CONUT scores.” However, PLR is completely absent from the Abstract and the Results tables (Tables 1-7). If the analysis of PLR was performed, the data should be included. However, if the data was not included, references to PLR should not appear in the Introduction and Discussion.
Author Response: We thank the reviewer for this important comment. PLR was not analyzed in this study, and its inclusion in the Introduction and Discussion resulted from a typographical error. We have corrected this throughout the manuscript to ensure consistency with the data presented in the Introduction and Discussion.
- Section 3.1., the text refers to “NLP”. Please clarify if NLP is a typo for another variable and if the value is correct.
Author Response: We thank the reviewer for carefully identifying this issue. The term “NLP” in Section 3.1. is a typographical error. There is no variable termed “NLP” in this study. The correct term is NLR (neutrophil-to-lymphocyte ratio), which is consistently used throughout the manuscript. The reported value corresponds to NLR and is correct. We have corrected this typo in the revised manuscript and clarified the full term at its first mention.
- Section 3.2., "Although LMR, PNI, and CONUT showed significance in univariate analysis.", This goes against the date in Tables 2 and 3. Based on the tables you provided, PNI was not significant. Please make the appropriate corrections regarding the data.
Author Response: We thank the reviewer for this important comment. We agree that the original statement in Section 3.2 was inconsistent with the data presented in Tables 2 and 3. As correctly noted, PNI was not statistically significant in the univariate analysis. We have revised the text accordingly to ensure full consistency with the presented data in line 270.
- The study uses median values as the cut-off points. While acceptable, the authors should have explained why they used the median score, when ROC curve analysis for the Youden index is usually the preferred method for getting the optimal cut-off points.
Author Response: We thank the reviewer for this thoughtful comment. We agree that ROC curve analysis using the Youden index is a commonly used approach to determine optimal cut-off values. In the present study, we deliberately used median values as cut-off points for the preoperative scores because this was a retrospective, exploratory prognostic analysis, and no universally accepted or validated cut-off values currently exist for these indices in patients with iCCA.
Using the median cut-off reduces the risk of outcome-driven optimization and overfitting, facilitates balanced group sizes, and improves reproducibility across different cohorts. ROC-derived cut-offs may be cohort-specific and ideally require independent external validation.
- Lines 448-448, “NLR demonstrated superior prognostic performance compared to ……. ,including the PLR." Since the PLR data is not shown in the results, this claim should be revised.
Author Response: We thank the reviewer for this comment. We agree that the original statement was unclear. The term “PLR” was used in error and should have been LMR (lymphocyte-to-monocyte ratio), which was the index analyzed and reported in the Results section. We have corrected this terminology throughout the manuscript to ensure consistency, and the statement has been revised accordingly.

Reviewer 3 Report
Comments and Suggestions for Authors
This study was aimed to investigate prognostic implication of inflammatory and immune-nutritional markers in patients with ICC who underwent surgical resection retrospectively. The number of patients was relatively enough as the single institutional data because authors institution has high-volume center of ICC over the world. However, there were some issues to be clarified in this manuscript at the present style.
1, Why did not authors include other preoperative factors in univariate and multi-variate analysis for predicting OS and DFS such as tumor factors of tumor size, tumor markers in this study? This study included only inflammatory and immune-nutritional indices without other tumor-related factors of ICC.
2, Authors addressed the clinical importance not only NLR but also CONUT indices. However, significant usefulness of CONUT was revealed only in the subgroup analysis of prognostic value in patients with recurrence and margin-negative curative resection.
CONUT could not achieve useful implications in all patients with ICC after surgical resection although NMR obtained strong prognostic value in these group.
3, Authors should speculate the scientific reasons why results in subgroup study did not match and show different significant implications between NMR and CONUT factors.
4, it has been already revealed that there were macroscopic patterns of ICC growth such as mass-forming type, periductal type, and intraductal type reflecting the clinical prognosis. It was uncertain why authors did not utilize these growth patterns evaluated with preoperative imaging such as CT, and MRI.
5, In this study post-recurrence survival was investigated in the subgroup analysis. However, it might be sometimes very difficult to estimate the time of recurrence clearly in most patients after surgical resection. How did authors determine the time of recurrence in this study?
6, Were there any therapeutic modalities such as chemotherapy, irradiation and redo surgery in recurrent patients in this study group? It should be clearly shown in this study.
Author Response
Overall response and summary of revisions
We sincerely thank the Editor and all reviewers for their constructive and insightful comments, which improved the clarity, methodological transparency, and clinical interpretation of our manuscript entitled “Preoperative Prognostic Score for Patients with Intrahepatic Cholangiocarcinoma Undergoing Curative-Intent Resection.” We have addressed all points and revised the manuscript accordingly. The major revisions include: (1) clarification of cohort definition and surgical intent, (2) correction of terminology and internal inconsistencies, (3) strengthening of the Methods and endpoint definitions, (4) improved presentation and consistency of Results and tables, and (5) refinement of the Discussion/Conclusion to avoid causal language and emphasize cohort-level risk stratification and appropriate clinical applicability. Detailed point-by-point responses are provided below.
Key revisions across the manuscript
- Terminology and cohort definition: Replaced “curative resection” with “curative-intent resection” throughout to reflect intent at resection (with margin and nodal status confirmed postoperatively). Clarified inclusion of R1 resections and resected nodal metastases when no gross residual disease remained, ensuring consistency across Abstract/Methods/Results.
- Methods and outcomes: Clarified that only iCCA cases were included and updated Figure 1 to show exclusions of pCCA/dCCA. Revised DFS definition to standard practice (time to recurrence or death from any cause) and removed the assumption that deaths without recurrence were cancer-related. Clarified recurrence ascertainment using routine surveillance and acknowledged that recurrence timing reflects first detection.
- Data consistency and corrections: Removed erroneous references to PLR (not analyzed), corrected “NLP” to “NLR,” and revised statements (univariate significance for PNI) to match the tables.
- Results presentation: Clarified reporting of age to avoid subgroup misinterpretation, revised BMI reporting (median and range), and added complete abbreviation definitions in table footnotes.
- Cut-offs and limitations: Justified the use of median-based cut-offs for NLR/LMR/PNI in the absence of validated iCCA thresholds, and explicitly noted that dichotomization supports cohort comparisons but limits individualized prognostic inference (with continuous-variable analyses addressed where applicable).
- Clinical interpretation: Revised Discussion/Conclusion to emphasize association (not causation) and that mechanistic confirmation requires dedicated studies. Clarified that CONUT findings were context-dependent (mainly in the R0 + recurrence subgroup) and that indices should be used as adjunctive tools for cohort-level risk stratification, not determinants of surgical eligibility.
- Limitations and future directions: Acknowledged missing/incomplete data (imaging tumor features, tumor markers, adjuvant/post-recurrence treatment, longitudinal indices) and highlighted the need for prospective studies with standardized data capture and evaluation of prehabilitation/nutrition protocols.
Reviewer 3
Open Review
(x) I would not like to sign my review report
( ) I would like to sign my review report
Quality of English Language
( ) The English could be improved to more clearly express the research.
(x) The English is fine and does not require any improvement.
|
Yes |
Can be improved |
Must be improved |
Not applicable |
|
|
Does the introduction provide sufficient background and include all relevant references? |
(x) |
( ) |
( ) |
( ) |
|
Is the research design appropriate? |
( ) |
( ) |
(x) |
( ) |
|
Are the methods adequately described? |
( ) |
( ) |
(x) |
( ) |
|
Are the results clearly presented? |
( ) |
(x) |
( ) |
( ) |
|
Are the conclusions supported by the results? |
( ) |
( ) |
(x) |
( ) |
|
Are all figures and tables clear and well-presented? |
( ) |
(x) |
( ) |
( ) |
Comments and Suggestions for Authors
This study was aimed to investigate prognostic implication of inflammatory and immune-nutritional markers in patients with ICC who underwent surgical resection retrospectively. The number of patients was relatively enough as the single institutional data because authors institution has high-volume center of ICC over the world. However, there were some issues to be clarified in this manuscript at the present style.
1, Why did not authors include other preoperative factors in univariate and multi-variate analysis for predicting OS and DFS such as tumor factors of tumor size, tumor markers in this study? This study included only inflammatory and immune-nutritional indices without other tumor-related factors of ICC.
Author Response Q1: We thank the reviewer for this important and insightful comment. We fully agree that tumor-related factors, including preoperative imaging–derived tumor characteristics (such as tumor size and tumor morphology) and serum tumor biomarkers, are clinically relevant predictors of OS and DFS in patients with iCCA.
In the present study, our analyses were deliberately focused on preoperative inflammatory and immune-nutritional indices, as these parameters were consistently available across the cohort. This study was retrospective in nature and included patients treated between 2015 and 2021. During the earlier years of this period, comprehensive preoperative imaging data and tumor biomarker measurements were not systematically or uniformly recorded, resulting in a high proportion of missing data—particularly for imaging-based tumor characteristics.
To avoid substantial case exclusion and potential selection bias, these tumor-related variables were not included in the univariate and multivariable models. We acknowledge this as an important limitation of the study. The reviewer’s suggestion is highly valuable, and we are currently improving our data collection and storage system to ensure more complete capture of preoperative imaging and tumor biomarker data. This will allow incorporation of tumor-related factors into integrated prognostic models in future prospective and updated cohort studies.
Discussion in line 599-605:
“Third, several clinically relevant preoperative variables, including imaging-based tumor characteristics (such as tumor size and macroscopic morphology) and serum tumor biomarkers, were incompletely recorded, precluding their inclusion in the multivariable Cox regression models. This limitation may have restricted evaluation of the independent prognostic contribution of tumor-related factors and may partly explain the predominant significance observed for inflammatory and nutritional indices.”
2, Authors addressed the clinical importance not only NLR but also CONUT indices. However, significant usefulness of CONUT was revealed only in the subgroup analysis of prognostic value in patients with recurrence and margin-negative curative resection.
CONUT could not achieve useful implications in all patients with ICC after surgical resection although NMR obtained strong prognostic value in these group.
Author Response Q2: Thank you for this important observation. We agree with the reviewer that the prognostic value of the CONUT score should not be overstated. In our study, CONUT did not demonstrate prognostic significance in the overall iCCA cohort. Its association with survival emerged only in specific subgroups, particularly among patients with recurrence and those who achieved R0 resection.
We have revised the Abstract, Discussion, and Conclusion to clarify that the prognostic relevance of CONUT appears to be context-dependent and limited to certain clinical subtypes rather than applicable to all patients undergoing curative-intent resection. The revised manuscript now emphasizes NLR as the more consistent and broadly applicable preoperative marker in this population.
Abstract
“The CONUT score was associated with OS in both R0 and recurrent subgroups. Tumor morphology, consistent with prior reports, was identified as a postoperative pathological factor associated with worse prognosis. Conclusion: Preoperative NLR was associated with poorer DFS and OS in iCCA patients undergoing curative-intent resection. This association was consistently observed in subgroups with R0 resection and in those who developed recurrence. Meanwhile, the CONUT score showed limited independent significance only among patients with R0 resection who experienced recurrence.”
Discussion
At line 456-465: “We observed that elevated preoperative NLR was consistently associated with poorer disease-free survival (DFS) and overall survival (OS), and that this association persisted across important clinical subsets, including patients who achieved margin-negative (R0) resection and those who subsequently developed recurrence. In contrast, the prognostic association of the CONUT score appeared more selective and context dependent, with independent significance observed only in the subgroup of patients with R0 resection who experienced recurrence. Collectively, these findings indicate that preoperative inflammatory and nutritional indices are associated with long-term outcomes in resected iCCA, with NLR demonstrating more consistent prognostic relevance across patient subgroups.”
At line 542-544: “Consequently, even after achieving R0 resection, patients with elevated CONUT scores who experienced recurrence demonstrated worse survival outcomes.”
At line 555-562: “Among R0-resected patients who experience recurrence, a high CONUT score may reflect reduced capacity to cope with tumor burden and impaired anti-tumor immune response, factors that are not fully represented by inflammatory markers alone [17,49]. Therefore, while NLR broadly predicts outcomes related to tumor biology and systemic inflammation, CONUT provides complementary prognostic information, particularly in recurrent cases, where the interplay between nutritional status, physiological reserve, and tumor aggressiveness critically influences survival.”
Conclusion
“This study demonstrates that elevated preoperative NLR is an independent and consistent associated with poorer DFS and OS in patients with iCCA undergoing curative-intent resection. The prognostic value of NLR persisted across clinically relevant subgroups, including patients who achieved margin-negative resection and those who subsequently developed recurrence, supporting its role as a broadly applicable marker of adverse tumor biology and systemic inflammatory status. In contrast, the impact of the CONUT score was limited and context dependent, with independent significance observed only in patients with margin-negative resection who later experienced recurrence, suggesting a selective role of nutritional status in specific clinical settings. Taken together, these findings indicate that preoperative systemic inflammation represents a central determinant of long-term prognosis, whereas nutritional impairment may exert additional prognostic influence only in selected patient subsets. Accordingly, NLR appears suitable for general preoperative risk stratification, while the CONUT score may provide supplementary information in specific clinical contexts.”
3, Authors should speculate the scientific reasons why results in subgroup study did not match and show different significant implications between NMR and CONUT factors.
Author Response Q3: We thank the reviewer for this important comment. We have added a detailed biological interpretation in the Discussion to explain the discrepant subgroup findings between NLR and CONUT. Briefly, NLR primarily reflects systemic inflammation and tumor–host interactions that influence disease behavior across all stages of iCCA, which may account for its consistent prognostic value across subgroups. In contrast, CONUT reflects nutritional status and physiological reserve, which are influenced by multiple host-related factors and may become prognostically relevant only in specific clinical contexts, such as patients with recurrence after R0 resection. These mechanistic considerations have been incorporated into the revised Discussion.
Discussion of NLR at line 515-531:
“These findings support the potential role of NLR as a prognostic biomarker reflecting aggressive tumor behavior and an unfavorable disease course in iCCA. NLR is widely regarded as an indicator of the balance between systemic inflammation and host immune response. An elevated NLR reflects a relative increase in circulating neutrophils and/or a reduction in lymphocytes, a profile that has been associated with tumor-promoting conditions and adverse clinical outcomes. Prior studies have shown that neutrophils can facilitate tumor progression through the secretion of pro-inflammatory cytokines, growth factors, and matrix-degrading enzymes, thereby promoting angiogenesis, invasion, and metastasis [38,39]. In addition, neutrophil-derived reactive oxygen species and arginase have been reported to suppress lymphocyte proliferation and cytotoxic activity, contributing to impaired anti-tumor immunity [40,41]. In contrast, lymphocytes, particularly cytotoxic T cells and natural killer cells, play a central role in tumor surveillance and immune-mediated tumor control. A reduced lymphocyte count, as reflected by a high NLR, has therefore been associated with compromised adaptive immunity and diminished tumor control [42]. Taken together, these biologically plausible mechanisms described in prior literature may help explain why elevated preoperative NLR has been consistently associated with poorer DFS and OS across multiple cancer types, including iCCA [18,30,36,43]”
Discussion of CONUT at line 539-562:
“Importantly, we found that a higher preoperative CONUT score, especially in the moderate to severe range, was associated with poorer prognosis. This likely reflects underlying malnutrition and weakened physical condition prior to resection, which could limit patients’ physiological reserves and impair immune responses [17,45-47]. Consequently, even after achieving R0 resection, patients with elevated CONUT scores who experienced recurrence demonstrated worse survival outcomes. In our study, the NLR was found to reflect systemic inflammation and immune status both key factors in tumor progression and metastasis. An elevated NLR indicates a pro-tumor inflammatory state and immunosuppression, which influence disease behavior across all stages of iCCA. This explains why NLR serves as a reliable prognostic marker for both DFS and OS, including in patients with margin-negative (R0) resections and those who develop recurrence. In contrast, the CONUT score reflects nutritional status and physiological reserve, incorporating serum albumin, total cholesterol, and lymphocyte count, which collectively indicate a patient’s ability to tolerate stress and maintain immune competence [48]. While systemic inflammation, as captured by NLR, is a key driver of tumor progression, adequate nutrition and physical robustness are essential for postoperative recovery, immune surveillance, and resistance to recurrent disease [43]. Among R0-resected patients who experience recurrence, a high CONUT score may reflect reduced capacity to cope with tumor burden and impaired anti-tumor immune response, factors that are not fully represented by inflammatory markers alone [17,49]. Therefore, while NLR broadly predicts outcomes related to tumor biology and systemic inflammation, CONUT provides complementary prognostic information, particularly in recurrent cases, where the interplay between nutritional status, physiological reserve, and tumor aggressiveness critically influences survival.”
4, it has been already revealed that there were macroscopic patterns of ICC growth such as mass-forming type, periductal type, and intraductal type reflecting the clinical prognosis. It was uncertain why authors did not utilize these growth patterns evaluated with preoperative imaging such as CT, and MRI.
Author Response Q4: We thank the reviewer for this important and insightful comment. We fully agree that macroscopic growth patterns of iCCA, including mass-forming, periductal infiltrating, and intraductal growth types, assessed by preoperative imaging such as CT and MRI, are clinically relevant and have been shown to be associated with prognosis. However, the present study was retrospective and included patients treated during earlier periods, in which standardized radiologic classification of macroscopic growth patterns was not routinely or consistently available for all cases. As a result, incorporation of preoperative imaging–based growth patterns would have led to substantial missing data and potential selection bias. Therefore, these imaging-derived variables were not included in the current analyses. In the present cohort, information on macroscopic growth patterns was available only from postoperative pathological assessment. Although postoperative pathological growth patterns are recognized prognostic factors for survival outcomes, they do not represent preoperative variables and therefore fall outside the primary scope of this study, which focused on preoperative risk stratification.
The primary objective of this study was to comparatively evaluate the prognostic performance of established preoperative inflammatory and nutritional indices derived from routinely measured laboratory parameters, which are commonly used as adjunctive tools in general clinical practice, rather than to develop a modified or novel scoring system. In addition, we aimed to examine their prognostic impact both in the overall cohort and within clinically relevant subgroups.
We acknowledge the absence of imaging-based growth patterns as an important limitation of the study. Future prospective studies with standardized preoperative imaging protocols may allow integration of imaging-derived macroscopic growth patterns with inflammatory and nutritional indices to develop more comprehensive preoperative prognostic models.
Discussion in line 599-605: “Third, several clinically relevant preoperative variables, including imaging-based tumor characteristics (such as tumor size and macroscopic morphology) and serum tumor biomarkers, were incompletely recorded, precluding their inclusion in the multivariable Cox regression models. This limitation may have restricted evaluation of the independent prognostic contribution of tumor-related factors and may partly explain the predominant significance observed for inflammatory and nutritional indices.”
5, In this study post-recurrence survival was investigated in the subgroup analysis. However, it might be sometimes very difficult to estimate the time of recurrence clearly in most patients after surgical resection. How did authors determine the time of recurrence in this study?
Author Response: We thank the reviewer for this important comment. In this study, the time of recurrence was determined based on routine postoperative surveillance, which included scheduled physical examinations and imaging assessments using computed tomography (CT) or magnetic resonance imaging (MRI) of the chest, abdomen, and pelvis. Follow-up evaluations were performed every three months during the first two years after resection, every six months during years three to five, and annually thereafter, with additional investigations conducted when clinically indicated. Recurrence was defined by radiologic evidence, histopathological confirmation when available, or documentation in the medical records as presented in method section: 2.5. Postoperative Follow-up and Outcome measurements.
We acknowledge that, due to the interval-based nature of routine surveillance, the exact timing of recurrence could not always be determined with absolute precision, particularly beyond the first two years after surgical resection. Accordingly, the recorded recurrence date reflects the time at which recurrence was first detected rather than the exact biological onset. We agree that this represents an important limitation of the study, which has now been explicitly acknowledged in the manuscript. Future prospective studies with standardized and more frequent assessments may allow more accurate determination of recurrence timing.
Discussion in limitation at line 610-612:
“Similarly, owing to the retrospective design and interval-based routine surveillance, some imprecision in determining the exact timing of recurrence was unavoidable, particularly during long-term follow-up.”
6, Were there any therapeutic modalities such as chemotherapy, irradiation and redo resection in recurrent patients in this study group? It should be clearly shown in this study.
Author Response: We thank the reviewer for this important and clinically relevant comment. This study was retrospective in nature, and information regarding therapeutic modalities after recurrence, including chemotherapy, radiotherapy, and redo resection, was incomplete and not consistently documented in the medical records across the study period. As a result, post-recurrence treatments could not be reliably categorized, quantified, or analyzed, and therefore were not included in the present study.
We agree that post-recurrence management may influence survival outcomes and acknowledge this as an important limitation of the study, which has now been explicitly stated in the manuscript. Future prospective studies with systematic and standardized collection of post-recurrence treatment data are warranted to more accurately assess the impact of subsequent therapies on post-recurrence survival.
Discussion in line 619-622:
“Finally, post-recurrence treatments, including chemotherapy, radiotherapy, and redo resection, were incompletely documented and could not be reliably categorized or analyzed, and thus were not included in the present study.”

Round 2
Reviewer 2 Report
Comments and Suggestions for Authors
I would like to thank the authors for the modifications and clarifications made throughout the manuscript. I believe the manuscript can be accepted for publication in its current form.
Author Response
Thank you very much for your kind comments and for recognizing the revisions made to the manuscript. We sincerely appreciate your time and thoughtful evaluation.
Reviewer 3 Report
Comments and Suggestions for Authors
Nothing to be required for any further revision.
Author Response
Thank you very much for your kind comment. We sincerely appreciate your time and consideration.